# Environmental Controls on Isolated Convection during the Amazonian Wet Season

Leandro Alex Moreira Viscardi[1,2], Giuseppe Torri[2], David K. Adams[3], and Henrique de Melo Jorge Barbosa[1,4]

[1]Institute of Physics, University of São Paulo, São Paulo, SP, Brazil
[2]Department of Atmospheric Sciences, University of Hawai'i at Mānoa, Honolulu, HI, USA
[3]El Instituto de Ciencias de la Atmósfera y Cambio Climático, Universidad Nacional Autónoma de México, Mexico City, Mexico
[4]Physics Department, University of Maryland Baltimore County, Baltimore, MD, USA

**Correspondence:** Leandro Viscardi (viscardi@hawaii.edu) and Henrique Barbosa (hbarbosa@umbc.edu)

**Abstract.**

   The Amazon rainforest is a vital component of the global climate system, influencing the hydrological cycle and tropical circulation. However, understanding and modeling the evolution of convection in this region remains a scientific challenge. Here, we assess the environmental conditions associated with shallow, congestus, and isolated deep convection days during the wet season (December to April) employing measurements from the GoAmazon (2014-2015) experiment and large-scale wind field from the constrained variational analysis. Composites of deep days show moister than average conditions below 3 km early in the morning. Analyzing the water budget at the surface through only observations, we estimated the water vapor convergence term as a residual of the water balance closure. Convergence remains nearly zero during the deep days until early afternoon (13 LST), when it becomes a dominant factor in their water budget. At 14 LST, the deep days experience a robust upward large-scale vertical velocity, especially above 4 km, which supports the shallow-to-deep convective transition occurring around 16-17 LST. In contrast, shallow and congestus days exhibit pre-convective drier conditions, along with diurnal water vapor divergence and large-scale subsidence that extend from the surface to the lower free troposphere. Moreover, afternoon precipitation exhibits the strongest linear correlation (0.6) with large-scale vertical velocity, nearly double the magnitude observed for other environmental factors, even moisture, at different levels and periods of the day. Precipitation also exhibits a moderate increase with low-level wind shear, while upper-level shear has a relatively minor negative impact on convection.

## 1 Introduction

In the tropics, deep convection dominates the weather and climate. The formation of cumulus clouds and their evolution into cumulonimbus, and often into mesoscale convective systems (MCS), covers a wide range of spatial and temporal scales. Similarly, complex physical processes from cloud microphysics to gravity wave generation are intrinsically tied to deep convection (Mapes et al., 2006; Mapes and Neale, 2011; Jewtoukoff et al., 2013; Gupta et al., 2023). Representing these convective cloud processes in general circulation models (GCMs) is recognized as a source of bias and uncertainty (Sherwood et al., 2014;

Stevens and Bony, 2013; Itterly et al., 2018; Maher et al., 2018; Freitas et al., 2020). In particular, the shallow-to-deep (STD) convective transition, whose physical mechanisms are still debated (Kuang and Bretherton, 2006; Khairoutdinov and Randall, 2006; Wu et al., 2009; Waite and Khouider, 2010; Schlemmer and Hohenegger, 2015; Morrison et al., 2022; Barber et al., 2022), has long been problematic for convective parameterizations (Betts, 2002; Betts and Jakob, 2002; Bechtold et al., 2004; Grabowski et al., 2006).

To evaluate model performance and to limit the uncertainty associated with the representation of deep convection either parameterized or resolved at the cloud-scale, metrics derived from observations are required (Adams et al., 2013, 2017; Schiro et al., 2018; Barber et al., 2022). However, within the deep tropics, data are typically lacking at the resolution needed to evaluate both parameterized convection in GCMs and cloud-resolving models (CRMs), as well as to investigate the physical processes that drive mesoscale convective evolution. Intensive field campaigns, such as the Green Ocean Amazon (GOAmazon) 2014/5 experiment (Martin et al., 2016, 2017), have provided comprehensive measurements from the surface to the clouds, finally providing critical measurements from microphysical scale to large-scale thermodynamic properties critical for studying convection (Giangrande et al., 2016, 2017, 2020; Feng and Giangrande, 2018; Barber et al., 2022).

Convective processes in the Amazon manifest themselves in two primary precipitation modes: isolated convection, which peaks in the late afternoon (around 16-17 LST), and organized convection associated with MCSs, which often peaks during the night into the early morning hours, around 03-04 LST. This distinct cycle is pronounced in the southern (Tota et al., 2000; Machado, 2002; Silva Dias, 2002) and central (Greco et al., 1990; Adams et al., 2013; Ghate and Kollias, 2016; Giangrande et al., 2020) regions. In particular, isolated convection is tied to the diurnal cycle of thermodynamic and dynamical properties (Ghate and Kollias, 2016; Itterly et al., 2018; Zhuang et al., 2017; Tian et al., 2021). For example, using 3.5 years of water vapor fields from the Amazon Dense Global Navigation Satellite System (GNSS) Meteorological Network, Adams et al. (2013) observed a 4-hr timescale of robust water vapor convergence associated with the STD transition, regardless of convective strength, seasonal variations, or whether convection occurs during day or night. For the vertical moisture distribution, Ghate and Kollias (2016) noted that precipitating days in the dry season start the diurnal cycle by exhibiting a moister layer around 2-5 km. However, this contrasts with later studies, which observed that deep convection days are associated with a moister layer from the surface to mid-levels ($\sim$ 5 km) (Zhuang et al., 2017; Tian et al., 2021).

Itterly et al. (2016) highlighted that convection in the Amazon region has a greater sensitivity to column water vapor, while CAPE turns out to be an inadequate indicator of convection intensity. Further support for these findings comes from the work of Schiro et al. (2016, 2018), who demonstrated the necessity of incorporating deep-inflow mixing to accurately represent the entrainment of dry air in the parcel, which is essential for assessing the sensitivity of convection to instability. Itterly et al. (2018) reaffirmed the importance of the vertical distribution of humidity for the evolution of convection, but they showed that the relationship between convection and humidity is represented very differently in reanalysis data products. Previous studies have not provided conclusive evidence for the importance of vertical wind shear. For example, Zhuang et al. (2017) proposed that stronger low- and upper-level bulk wind shear predominantly supports the STD transition, but specifically during the dry season (June-September). In contrast, Chakraborty et al. (2018) proposed that increased low-level shear could hinder deep convection, particularly from August to November, especially if it leads to greater entrainment of dry air.

Although the above studies have used a combination of observational and reanalysis data to investigate different aspects of convection, they have not reached a consensus on which variables are most strongly associated with convective triggering or intensity in tropical regions. Here, we investigate the STD transition employing data from the GoAmazon2014/5 rainy season observations. We analyze the diurnal cycle of days designated as shallow, congestus, and deep convection in terms of moisture, instability, and large-scale wind properties to assess the correlations of isolated convection in the Amazon to different environmental factors. In addition, we assess the correlation between conditionally averaged precipitation and these environmental factors to further determine their influence on convective intensity.

This paper is organized as follows: First, we present an overview of the GoAmazon2014/5 data and the procedures for convective regime classification are described in Section 2. A comparison of cloud and precipitation properties among the shallow, congestus, and deep days is given in Section 3. Section 4 describes the diurnal cycle of the cloud regimes, while Section 5 presents the analysis of conditionally averaged precipitation based on different environmental controls. A discussion is provided in section 6, and the conclusions are given in section 7.

## 2 Data and Methods

From January 2014 to December 2015, the GoAmazon2014/5 (Martin et al., 2016, 2017) experiment conducted the first long-term measurements of aerosols, clouds, and atmospheric thermodynamics in the central Amazon region. Nine ground sites and two aircraft were used to examine the environment in and around Manaus, a city that borders the Rio Negro and surrounded by tropical rainforest. A detailed description of all sites can be found in Martin et al. (2016). This study focuses on the region around the most instrumented site, T3, located 70 km downwind of Manaus, in Manacapuru (3.21°S, 60.60°W), where the AMF-1 and the ARM Mobile Aerosol Observing System (MAOS) were deployed.

### 2.1 Experimental and large-scale data

To begin with our convective regime classification, described in the next section, we employed a cloud mask based on time-height profiles of the cloud location derived from the Merged RWP-WACR-ARSCL Cloud Mask dataset which combines the Radar Wind Profiler (RWP) data with the original W-band Cloud Radar (WACR) Active Remote Sensing of Cloud (ARSCL) VAP both located at T3 (Giangrande et al., 2017; Feng and Giangrande, 2018). This cloud mask identifies seven types of clouds, three of which correspond to convective clouds: shallow cumulus, with a base and cloud top below 3 km; congestus, with a base below 3 km and a top between 3-8 km; and deep convection, with a base below 3 km and a top $> 8$ km. From the cloud mask, available every 30 s, we also compute the cloud frequency profile in 12-minute windows to match the S-band radar data described in the following.

The Amazon Protection System in Brazil (Sistema de Proteção da Amazônia, SIPAM), located at the T1 site (3.15°S, 59.99°W), operates an S-band (10 cm), Doppler, single-polarization scanning radar south of downtown Manaus. We used the 12-minute gridded reflectivity product with a horizontal resolution of 2 km and a vertical resolution of 500 m (Schumacher and Funk, 2018a). The S-band domain covers an area with a radius of 202 km. We use reflectivity profiles to assess rain coverage

in our analysis domain, 100x100 km² centered at T3 site, calculated using a reflectivity threshold of 20 dBZ as in Zhuang et al. (2017). The surface precipitation rate is derived from the reflectivity of a constant altitude plan position indicator (CAPPI) at 2.5 km, calibrated with surface measurements from a Joss-Waldvogel disdrometer at the T3 site during the wet season of early 2014 (Schumacher and Funk, 2018b). In examining conditionally averaged precipitation (Section 5), the precipitation rate is averaged over the analysis domain, along with an additional time-average from 14 to 20 LST.

Figure 1 shows the map of S-band precipitation averaged over the wet season (December to April) 2014-2015. The precipitation statistics indicate that some radar beams are partially blocked. We used a threshold of 4 mm day$^{-1}$ to roughly identify these problematic regions. Beam blockage affects only 7 out of 2,601 pixels in our analysis domain, corresponding to only 0.3%, with no relevant sensitivity to the chosen threshold. Although this contribution is negligible, we removed these blocked pixels from our analysis.

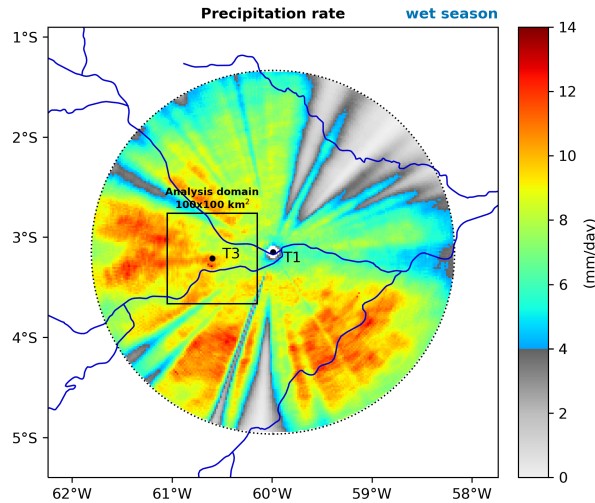

**Figure 1.** Map of precipitation rate at 2.5 km height averaged over the wet season (December to April) 2014-2015. The square box represents the analysis domain covering an area of 100x100 km² centered at the T3 site. The dotted circle (radius of 202 km) centered over the T1 site indicates the domain covered by S-band radar measurements.

In addition to rain cover, the analysis of cloud and precipitation properties (section 3) also uses the Aerosol Observing System (AOSMET) surface precipitation measured by the acoustic gauge of a Vaisala WXT520 station (ARM, 2013). A 12-minute average is also applied to match the cloud frequency and S-band radar data.

The balloon-borne sounding system (SONDE) provides the vertical profiles of thermodynamic properties four times per day, at 02, 08, 14, and 20 local standard time (LST) (ARM, 2014f) during the wet season (December-April). For consistency, we analyze the atmospheric profile (potential temperature and humidity) only for soundings that extend from the surface to at least 8 km, and we linearly interpolate the profiles to a fixed 50 m vertical grid. The planetary boundary layer height is derived from the profile of the bulk Richardson number using a critical threshold of 0.25 (ARM, 2014e). We also calculate additional variables from the thermodynamic profiles. The total column water vapor (CWV) is determined by integrating the

water vapor mixing ratio from the surface to 350 hPa. For consistency, here we only analyze soundings that extend from the surface to at least 350 hPa (approximately 8.5 km). Similarly, the partially-integrated CWV is calculated for the layers 1000-850 hPa ($\sim$ 0-1.5 km, $CWV_{lower}$), 850-700 hPa ($\sim$ 1.5-3 km, $CWV_{mid}$), and 700-350 hPa ($\sim$ 3-8 km, $CWV_{upper}$), which are representative of the convective boundary layer, lower free troposphere, and upper levels, respectively. Convective Inhibition (CIN) and Convective Available Potential Energy (CAPE) are also calculated. For these, a hypothetical air parcel is lifted dry-adiabatically to the Lifting Condensation Level (LCL), then pseudo-adiabatically from there. A mixed-layer parcel immediately above the surface, extending to a depth of 100 hPa, is used as the initial state for the parcel's ascent Stull (2016).

We used a combination of instruments to characterize the surface water budget. The latent heat flux is taken from the ARM best-estimate dataset (ARM, 2014a) based on observations from the Eddy Correlation (ECOR) Flux Measurement (ARM, 2014b). Instead of using the S-band radar precipitation, the water balance analysis utilizes a combination of rain-gauge source measurements to provide a more robust estimation of the mean surface precipitation and its uncertainties. Specifically, we use the surface AOSMET precipitation (ARM, 2013), tipping bucket (ARM, 2014g), and a laser disdrometer (ARM, 2014c). Here, CWV and liquid water path (LWP) are taken from the microwave radiometer (MWR, ARM, 2014d). Similarly to Schiro et al. (2016), we exclude data in cases where the brightness temperature surpasses 100 K, and when water accumulates on the MWR lens surface during rainy periods. Hourly averages are applied for the data utilized in the water budget analysis, and its mathematical derivation is provided in the next section.

Large-scale wind properties and bulk vertical wind shear are analyzed by way of Constrained Variational Analysis (VARANAL) data developed for the GoAmazon2014/5 experiment (Tang et al., 2016). This large-scale forcing dataset is derived from the European Centre for Medium-Range Weather Forecasts (ECMWF) analysis that is further constrained by S-Band radar precipitation, surface fluxes, and Geostationary Operational Environmental Satellite (GOES-13) radiances. The VARANAL large-scale fields represent an average across an analysis domain centered at the S-band radar location, with a radius of 110 km.

Finally, a table summarizing the data is also provided in the supplementary material (table A1).

## 2.2 Derivation of water budget

The integral form of the continuity equation for the total water $r_t$ can be written as,

$$\frac{\partial}{\partial t} \int_{p_{\text{top}}}^{p_{\text{bottom}}} r_t \frac{dp}{g} + \nabla \cdot \int_{p_{\text{top}}}^{p_{\text{bottom}}} r_t \boldsymbol{V} \frac{dp}{g} = E - P, \tag{1}$$

where $p$ is pressure, $g$ is gravity acceleration, $\boldsymbol{V}$ is the horizontal wind vector, and $E$ and $P$ correspond to the water mass fluxes associated with surface evaporation and precipitation. The total water $r_t$ is the sum of the three different phased-based mixing ratios, i.e., water vapor ($r_v$), liquid ($r_l$), and ice ($r_i$). The second integral is the water mass flux divergence, which is mostly associated with the divergence of water vapor.

For the sake of this analysis, we neglect ice and express all terms in units of mm hr$^{-1}$. As we show later (section 4.3), the results indicate a minor contribution of the liquid water term in the water budget, which supports ignoring the ice water

term. Note that ice water paths are not necessarily smaller than liquid water paths; however, they still encompass values of comparable orders of magnitude. Thus, the equation can be rewritten in a simplified form:

$$\frac{\partial}{\partial t}\mathrm{CWV} + \frac{\partial}{\partial t}\mathrm{LWP} - \mathrm{EVAP} + \mathrm{PREC} = \mathrm{CONV}, \tag{2}$$

where EVAP corresponds to the evaporation rate, PREC corresponds to the precipitation rate, and CONV represents the water vapor convergence. Note that we intentionally rearranged the order of the equation to emphasize terms based on GOAmazon observations on the left-hand side and the residuals, which can only be estimated, on the right-hand side. For this analysis, we consider only the hourly average of observed variables. Specifically, CWV and LWP are based on the MWR. ECOR latent heat flux data is utilized for estimating the evaporation term. Precipitation is obtained utilizing different sources, namely, aerosol observing system data, tipping bucket, and laser disdrometer (Section 2.1). We determine the mean of these sources and calculate the standard deviation from this sample mean. The water vapor convergence term is estimated using the mean composites for $\partial\mathrm{CWV}/\partial t$, $\partial\mathrm{LWP}/\partial t$, EVAP, and PREC. The standard deviation of mean convergence is estimated from the standard deviation of the mean $\partial\mathrm{CWV}/\partial t$, $\partial\mathrm{LWP}/\partial t$, EVAP, and PREC, i.e., $\sqrt{\sigma_m(\partial_t\mathrm{CWV})^2 + \sigma_m(\partial_t\mathrm{LWP})^2 + \sigma_m(\mathrm{EVAP})^2 + \sigma_m(\mathrm{PREC})^2}$. Although this formula involves implicit assumptions, such as a lack of correlation among the variables in the square root, the uncertainty in the water budget is primarily attributed to $\partial\mathrm{CWV}/\partial t$ (see fig. 9). Consequently, the uncertainty in the convergence term can also be roughly approximated by this term, with other terms contributing minimally.

Note that we only employed local surface measurements and applied hourly averages for the water budget datasets. Hence, the water balance is consistent over a temporal scale of an hour and a spatial scale on the order of meters. However, surface fluxes depend primarily on surface properties, which are approximately uniform around the experimental site, at least up to a distance of a few kilometers. Measurements of precipitation, CWV, and LWP are influenced by the cloud cover around the instrumentation location. Given that shallow-to-deep convection typically spans a spatial scale of approximately 1-10 km, our analysis is likely to generalize well across a spatiotemporal scale of one hour and a few kilometers.

## 2.3 Convective regime classification

We define the wet season as the period from December to April following previous studies in the Central Amazon (Machado et al., 2004; Marengo et al., 2013). To contrast the different atmospheric conditions that lead to different convective regimes, we apply a classification criterion that identifies shallow, congestus, or deep days. Given our interest in convection that develops in response to the diurnal cycle, days with propagating mesoscale convective systems (MCS) were excluded from our analysis. This regime classification follows the criteria proposed by Zhuang et al. (2017) and Tian et al. (2021). We first consider the diurnal period between 10-20 LST. Then, we define three categories of convective regimes: shallow cumulus, congestus, and deep, based on the maximum development of convective clouds during the diurnal cycle. The cloud mask from the RWP-WACR-ARSCL is used to identify cloud development throughout each day. In addition, we also employ rain coverage data from the S-band radar to estimate the vertical cloud development based on the echo cloud top. The rain coverage is calculated as the fraction of reflectivity pixels $> 20$ dBZ, regardless of whether the phase is ice or liquid (see Section 2.1). The echo-top is defined as the highest level where rain coverage is greater than 2%. The quantitative criteria are listed below:

i) Shallow convective days (ShCu): (1) The cloud mask does not show any congestus and deep clouds during the diurnal cycle (10-20 LST). (2) Rain cover above 3 km of altitude must be $< 2\%$ in the analysis domain (100x100 km$^2$, centered at T3).

ii) Congestus convective days (Cong): (1) The cloud mask indicates congestus clouds during the diurnal cycle, but none of them develops into deep clouds. (2) Rain cover above 8 km (between 3-8 km) of altitude must be $< 2\%$ ($> 2\%$) in the analysis domain.

iii) Deep convective days (Deep): (1) The cloud mask indicates deep convection during the diurnal cycle. (2) Rain cover above 8 km $> 2\%$ in the analysis domain.

iv) Early morning perturbation: For the 06-10 LST period, we require that no congestus and deep clouds be observed.

v) Local convection: No convective system with a contiguous area of precipitation $> 10,000$ km$^2$ reaches the analysis domain between 06-20 LST.

Figures 2 and 3 illustrate the diurnal evolution of the occurrence of the different cloud types and rain coverage for the three regimes. The classification essentially depends on the cloud evolution during the diurnal cycle. The threshold of 2% used to calculate rain cover is based on Zhuang et al. (2017), who manually tested several parameters. Their results indicated that shallow rain cover never exceeds 2%, a criterion we adopted in our definition of minimum rain cover for identifying congestus or deep clouds. The exclusion of any relevant early morning disturbance (06-10 LST) associated with important pre-convective activity guarantees that nighttime MCSs do not cause significant preconditioning.

The number of days categorized according to the above criteria for the wet season (blue) and dry season (red) is shown in Fig. 4. Deep convection appears to be the dominant category in both seasons, although the propagating-convection category occurs more frequently during the wet season. Specifically, we identified 16 days for the ShCu regime, 27 days for the Cong regime, 60 days for the Deep regime, and 111 days for non-local (organized) deep convection. Given our focus on the local shallow-to-deep transition mechanism during the wet season, the results presented in the next section refer only to those 103 days.

## 3   Cloud and Precipitation Properties

The composite diurnal cycles of the vertical cloud frequency profiles, local surface precipitation rate, and rain coverage for the different convective regimes are shown in Fig. 5. For all convective regimes, daytime convection is usually preceded by some nighttime cloud cover at all levels.

The ShCu regime has a more scattered cloud frequency during the daytime. Low-level clouds dominate the diurnal cycle, with a peak reaching 54% at 1.43 km around 13 LST. After 17 LST, cirrus clouds also contribute to the cloud frequency composite, likely transported from afar. The local surface precipitation rate remains below 0.13 mm hr$^{-1}$ throughout the diurnal cycle, while the rain cover has only a minor contribution of approximately 1% at the end of the day.

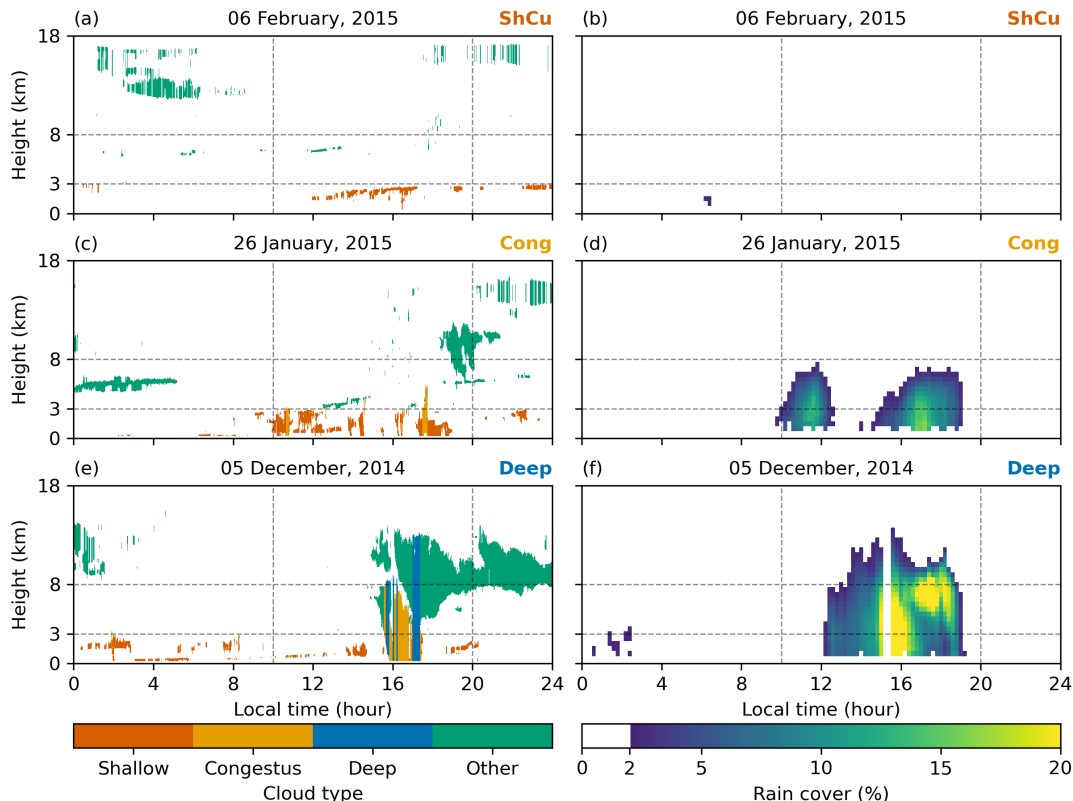

**Figure 2.** Cloud mask (left) and rain coverage (right) for examples of days classified as shallow (ShCu, a-b), congestus (Cong, c-d), and deep (Deep, e-f).

The Cong regime also shows a higher cloud frequency below 3 km during the diurnal cycle. The maximum cloud cover is 50% at 1.04 km around 12 LST, earlier and lower than observed for the ShCu regime. Between 3-8 km, the cloud frequency composite has values below 5%. Surface precipitation occurs around 12 LST, corresponding temporally with the period of maximum cloud cover. Nonetheless, the magnitude remains modest, with precipitation values at less than 0.75 mm hr$^{-1}$ throughout the diurnal cycle. Contrary to the local cloud frequency assessed at the T3 site, the domain rain coverage exhibits a distinct pattern at mid-levels, indicating values of 1-2%. After the diurnal cycle, the rain coverage composite indicates that congestus clouds may evolve into a deeper phase. We identified 7 cases (26%) in which congestus days fulfilled the Deep criteria (echo top above 8 km) after the diurnal cycle.

The Deep regime reveals a more coherent cloud frequency profile associated with more extensive and longer-lived convective clouds during daylight hours. The maximum Deep cloud frequency is also associated with shallow clouds, being 49% at 0.92 km around 11:30 LST. As we discuss next, this is associated with both the boundary layer height and the lifting condensation levels being lower in the Deep regime. As the day progresses, the Deep cloud frequency also increases throughout the troposphere, with cloud top reaching up to 16 km. After cumulonimbus dissipation (around 18 LST), its anvil structure remains

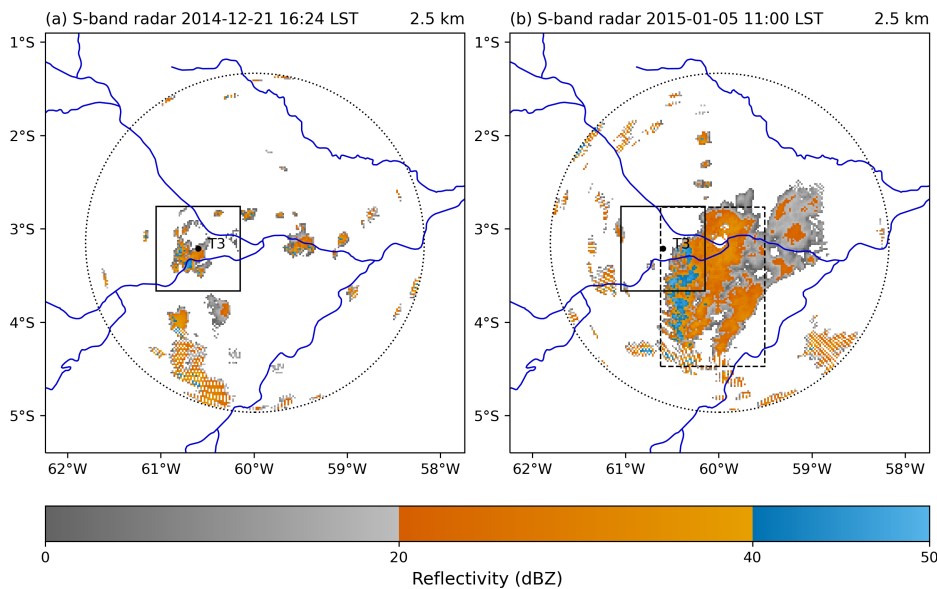

**Figure 3.** Convective system. (a) Scattered local system. (b) Non-local propagating system. The dashed black box illustrates the region with a contiguous area of precipitation ($> 20$ dBZ) not fulfilling the local convection requirement.

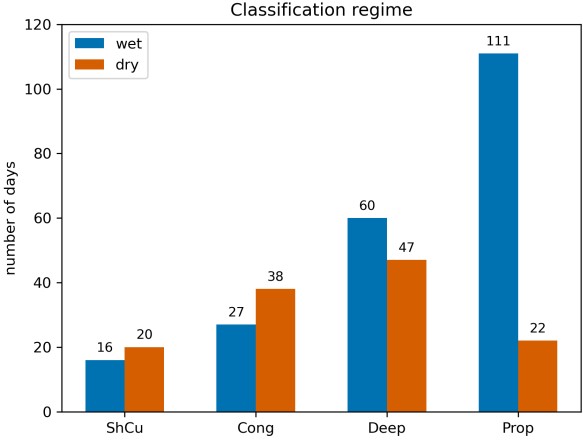

**Figure 4.** Number of days classified in each convective regime during the wet (Dec-Apr) and the dry (Jun-Aug) seasons, from 2014 to 2015. Propagating (Prop) days refer to non-local deep convection, with the early morning perturbation condition being ignored.

and may become a cirrus cloud that may contribute to the cloud cover of the next day. Surface precipitation typically occurs around noon and in the afternoon as clouds develop. Notably, a substantial rain coverage emerges between 16-17 LST, with the

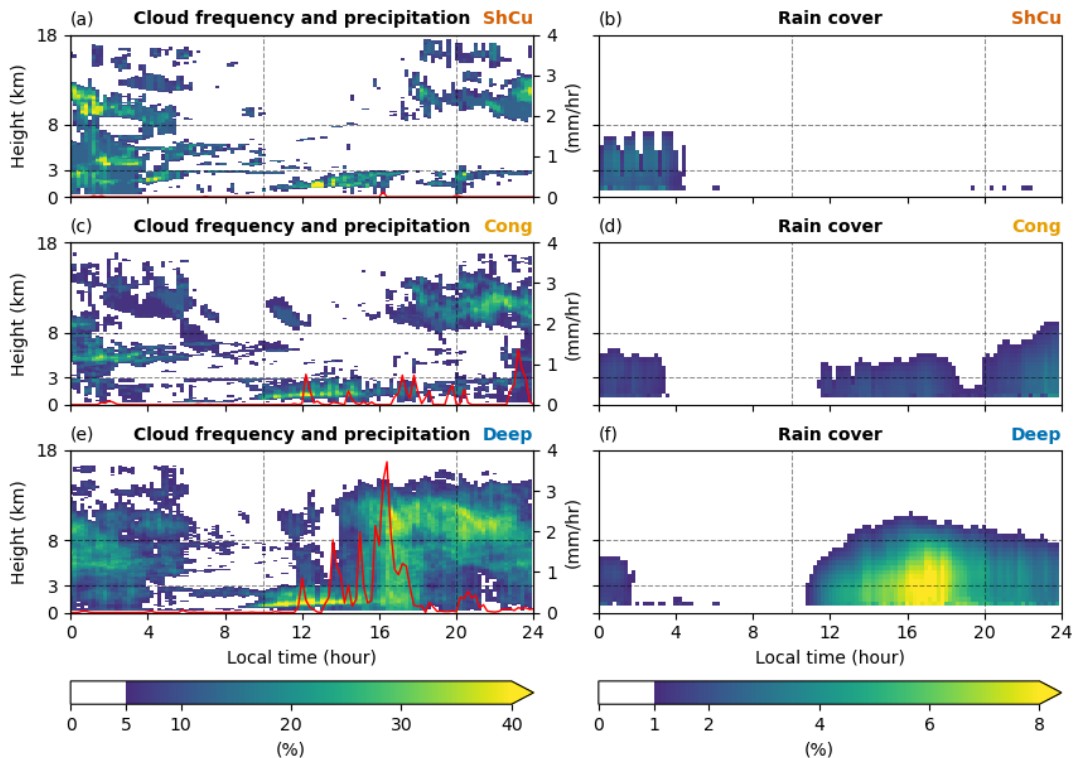

**Figure 5.** Left: Cloud frequency (cloud counting fraction in %, colormap) as a function of height calculated over 12 minutes window based on the cloud mask and surface precipitation rate (mm/hr, red line) locally measured by the aerosol observing system at the T3 site. Right: Rain cover (%) over the analysis domain (100x100 km$^2$, centered at T3) based on the S-band radar. The mean composites distinguish days classified as shallow (a-b, N=16), congestus (c-d, N=27), and deep (e-f, N=60) during the wet season.

maximum surface precipitation of 3.72 mm hr$^{-1}$ peaking at 16:24 LST. This occurrence is consistent with the late afternoon
STD transition in the Amazonian wet season.

## 4   Environmental Conditions

In this section, we specifically evaluate the environmental conditions associated with the STD convective transition. We analyze the local atmospheric conditions, convective properties, surface water balance, and large-scale wind properties associated with shallow, congestus, and deep convective days. To account for the small sample size associated with the classification of the
convective regime (fig. 4), we employed the bootstrap method, utilizing 50,000 samples to estimate the mean and standard deviation for each composite. These are represented by the line and error bars in each figure in this section.

## 4.1 Atmospheric Conditions

Figure 6 shows the average vertical profiles of potential temperature (Fig. 6a), $\theta$, and water vapor mixing ratio (Fig. 6b), $r_v$, for each convective regime at 08 LST. The differences between the Deep and other convective regimes, $\Delta\theta$ and $\Delta r_v$, are shown in panels c-d.

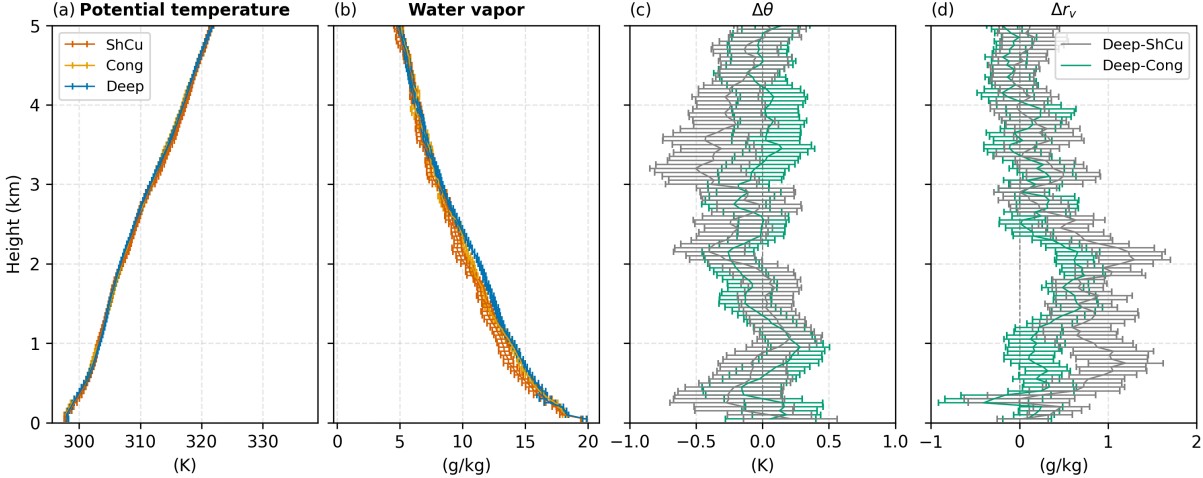

**Figure 6.** Atmospheric conditions. (a) Potential temperature (K) and (b) water vapor mixing ratio (g/kg) at 08 LST radiosonde observations. The corresponding convective regime differences (Deep-ShCu and Deep-Cong) for potential temperature ($\Delta\theta$, c) and mixing ratio ($\Delta r_v$, d) are also included. The error bars on panels c and d are obtained from the bootstrap standard error of the convective regimes, i.e., $\sqrt{\mathrm{SE(Deep)}^2 + \mathrm{SE(ShCu\ or\ Cong)}^2}$.

The potential temperature differences between the convective regimes are frequently less than 0.5 K below the 8 km level, suggesting no relevant relationship between daytime cloud development and the morning temperature profile. The Deep regime experiences higher moisture content, especially below 3 km. The most remarkable difference occurs around the 2 km level, with $\Delta r_v$ reaching 1.30 g kg$^{-1}$ and 0.74 g kg$^{-1}$ for Deep-ShCu and Deep-Cong, respectively. Above 3 km, the differences in moisture profile for all regimes are minimal.

These results suggest that the importance of early morning excess humidity to the STD transition in the Amazon is primarily limited to the lower levels, a finding supported by Zhuang et al. (2017); Tian et al. (2021). However, it differs from Ghate and Kollias (2016), which noted an excess of humidity solely above 2 km during precipitating days in the dry season. Moreover, these results contrast with studies over tropical oceans, where free-tropospheric humidity has been shown to play a more significant role in convective activity, while boundary layer humidity demonstrates minor variability (Bretherton et al., 2004; Holloway and Neelin, 2009).

The column water vapor (CWV) for each radiosonde time is presented in Fig. 7a, while panels b-d show the partial contribution from lower, middle, and upper levels (see Section 2.1). The ShCu regime shows the smaller CWV throughout the day. The difference between Deep and ShCu ranges from 2.3 mm at night to 5.1 mm at 14 LST. The differences in CWV among

245 Cong and Deep regimes show little statistical significance from nighttime to early morning, while their difference is maximum at 14 LST when it reaches 2.1 mm.

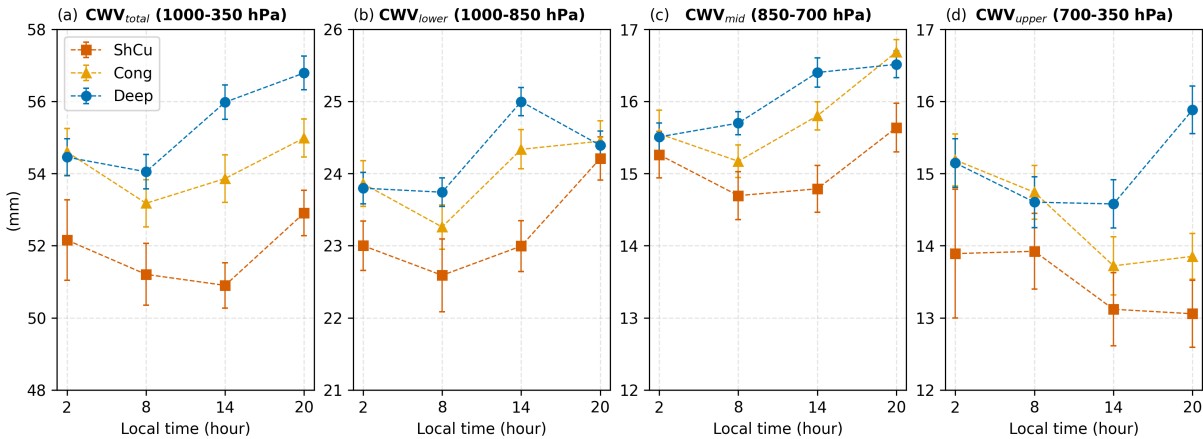

**Figure 7.** Column water vapor (CWV, mm) at the radiosonde launch times (02, 08, 14, and 20 LST). (a) Total and its partial contribution in the (b) 1000-850 hPa, (c) 850-700 hPa, and (d) 700-350 hPa layers.

The CWV$_{lower}$ increases from 08 LST to 14 LST for all convective regimes. This increase is possibly due to evapotranspiration, which appears to be the dominant moisture factor during this period (water budget analysis, Section 4.3). A similar, albeit less pronounced diurnal moistening is also observed in CWV$_{mid}$ for all regimes. However, the upper-level CWV is essentially

constant for the Deep regime in the 08-14 LST period, while the ShCu and Cong regimes show a decrease of 0.80 and 1.02 mm, respectively. This indicates that the ShCu and Cong regimes might be associated with large-scale subsidence (see Section 4.4), which would explain the drying of the mid-levels. Moreover, we note that Deep days show a significant increase in CWV$_{upper}$ from 14 to 20 LST, with a simultaneous decrease of CWV$_{lower}$. This is likely due to a combination of water vapor convergence (section 4.3) preceding the late afternoon STD transition and the vertical transport of moisture by deep clouds.

### 255   4.2   Convective Properties

Figure 8 shows the PBL height and the MLCIN and MLCAPE. The PBL height among the convective regimes differs the most at 14 LST, where the convective boundary layer roughly coincides with the LCL or cloud base height (not shown), being about 500 m lower for the Deep (1535 m) than the ShCu (1998 m) regime. The most developed convective regimes show a combination of higher MLCAPE and lower MLCIN in the early morning, providing more buoyancy and a lower barrier for

convection to develop. At 14 LST, MLCAPE for the Deep regime (1074 J kg$^{-1}$) and Cong regime (986 J kg$^{-1}$) significantly exceed the value for the ShCu regime (558 J kg$^{-1}$). The change in MLCAPE from afternoon to early evening (MLCAPE(20 LST) - MLCAPE(14 LST)) is negative for the Deep (-138 J kg$^{-1}$), and positive for the Cong (139 J kg$^{-1}$) and ShCu (644 J kg$^{-1}$) regimes. As expected, convection in more advanced stages consumes MLCAPE more effectively, reducing atmospheric instability.

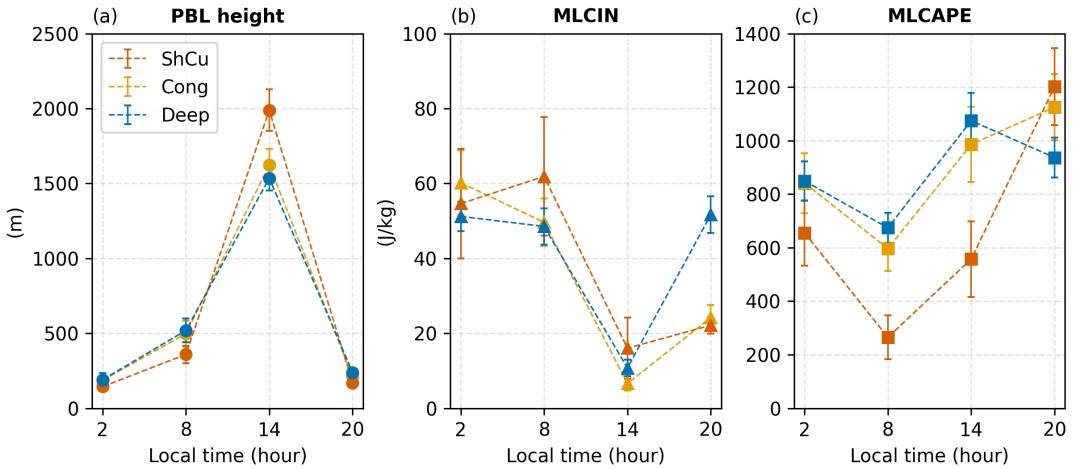

**Figure 8.** (a) Planetary boundary layer (PBL) height (m), (b) 100-hPa mixed-layer convective inhibition (MLCIN, J/kg), and (c) 100-hPa mixed-layer convective available potential energy (MLCAPE, J/kg) at the radiosonde launch times (02, 08, 14, and 20 LST).

### 4.3 Surface Water Balance

To understand how the differences in the surface fluxes change CWV, affecting the STD transition and the accumulated surface precipitation, we analyzed the surface water balance (see Section 2.2).

The water balance results are shown in Fig. 9, with the top panels showing the hourly average rate values (mm/day) and the bottom panels showing the corresponding accumulated values (mm). The separate precipitation and convergence associated with each rain gauge used to estimate the mean surface precipitation and its uncertainty in the water budget results are provided in fig. A1. First, we notice that the $\partial_t$LWP appears negligible, and it does not contribute significantly to the water budget. The daytime of ShCu and Cong days shows mostly water vapor divergence. This is primarily due to more significant negative changes in $\partial_t$CWV and low precipitation rates, as evaporation shows smaller differences among the convective regimes and changes in $\partial_t$LWP exert a minor influence on the water budget. On the other hand, the Deep regime exhibits relatively neutral conditions from night to early afternoon. However, the water budget closure requires convergence in the period from 14 to 18 LST. This coincides with the STD transition and is thus primarily attributed to strong water vapor convergence preceding the late afternoon STD transition (Adams et al., 2013, 2017). At day's end, the ShCu regime is estimated to lose 3.4 mm of water vapor due to divergence, while the Cong regime loses 0.9 mm. In contrast, the Deep regime gains 5.2 mm of water vapor due to convergence.

For ShCu days, relatively high surface evaporation and the absence of precipitation require a strong divergence for water balance closure. The Cong regime shows a significant divergence but is relatively weaker compared to the ShCu regime, as their surface evaporation is relatively similar, and congestus precipitation exhibits modest values. For ShCu days, the $\partial_t$CWV tends to be small and negative from nighttime to early morning. Then, it increases and balances around noon. By the end of the day, the accumulation of water vapor (fig. 9d) in the column is negligible. This term is also nearly zero during Cong

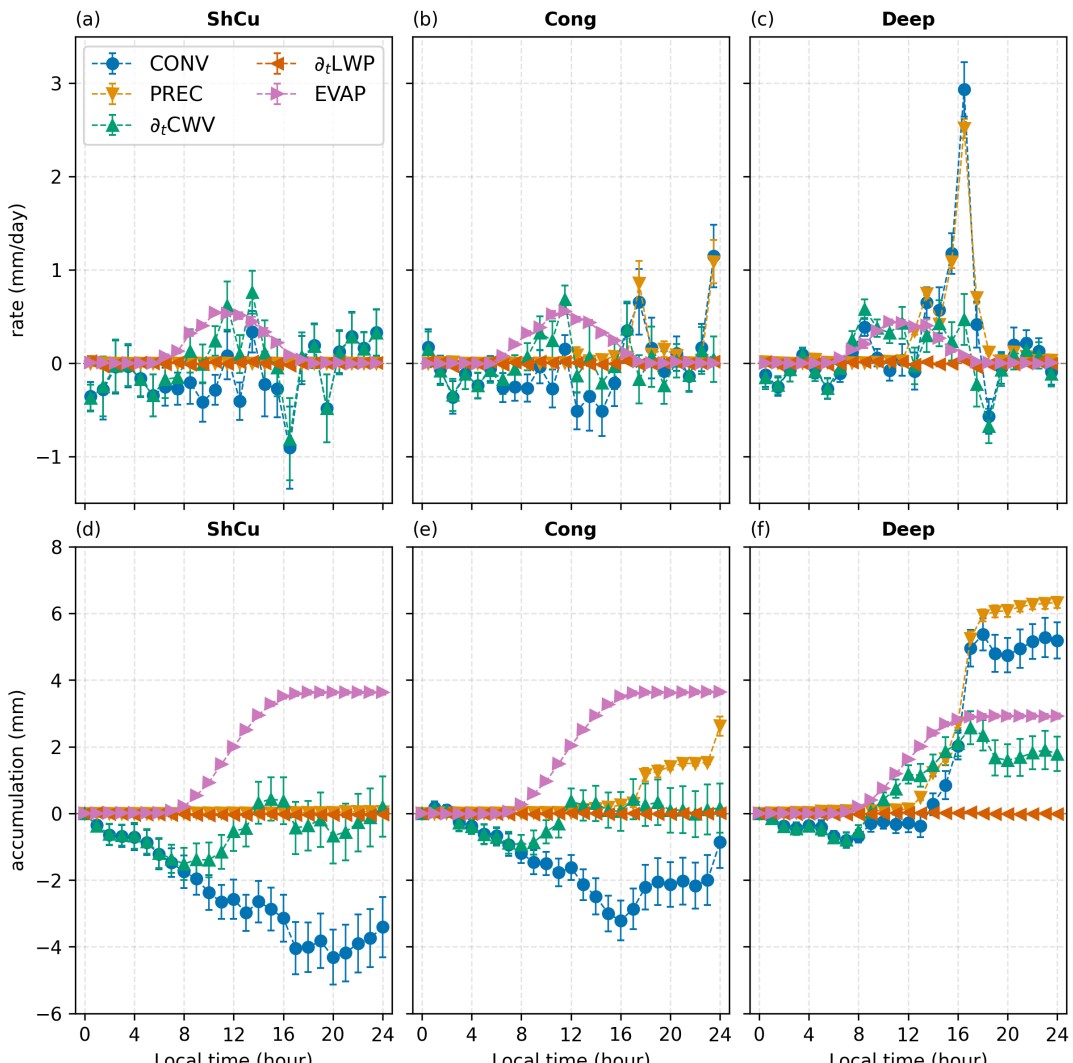

**Figure 9.** Surface water balance for (a,d) ShCu regime, (b,e) Cong regime, and (c,f) Deep regime. The upper panels represent the rates of change (mm/day) for water vapor convergence (CONV), evaporation (EVAP), precipitation (PREC), and the time derivatives of column water vapor ($\partial_t$CWV, where $\partial_t = \partial/\partial t$) and liquid water path ($\partial_t$LWP). The lower panels display the accumulated water amounts for each term in the water budget along the day (mm). Note that CWV and LWP changes rely on microwave radiometer observations, evaporation on eddy correlation flux measurements, and precipitation is estimated utilizing different sources, namely, the aerosol observing system surface data, tipping bucket, and laser disdrometer. The water vapor convergence term is estimated as a residual in the water budget equation (2).

285    days. Conversely, the convergence of vapor and evaporation exceeds the precipitation term on Deep days, resulting in a net accumulation of 1.8 mm in column water vapor, which might act as a positive feedback for the continuation of nocturnal deep

convection into the following day after Deep regime conditions. This particular finding had not been previously addressed in studies utilizing the GoAmazon2014/5 observations.

## 4.4   Large-Scale Wind Properties

Figure 10 displays the wind speed at 08 LST, 11 LST, and 14 LST for all convective regimes. The wind profiles for all regimes peak between the 900-800 hPa layer, characteristic of a low-level jet also reported by Anselmo et al. (2020), which they observed 10-40% of the time during GoAmazon between March-May 2014-2015. During the morning, the ShCu regime reveals a lower and slightly stronger jet. However, the PBL grows during the day to a height of 1-2 km (see Fig. 8), reaching higher altitudes for ShCu days, and the mixing of free-tropospheric and PBL momentum potentially reduces the wind speed.
As a result, the lower jet in the ShCu regime is likely more influenced by the PBL growth compared to other regimes. Thus, at 14 LST, the ShCu regime reveals a less prominent low-level wind peak but at a slightly higher altitude compared to Deep days. For the mid- and upper-levels, between 600-350 hPa layer, the ShCu regime shows an additional upper-level maxima, while the Cong and Deep regimes exhibit weaker and comparable wind speeds, respectively.

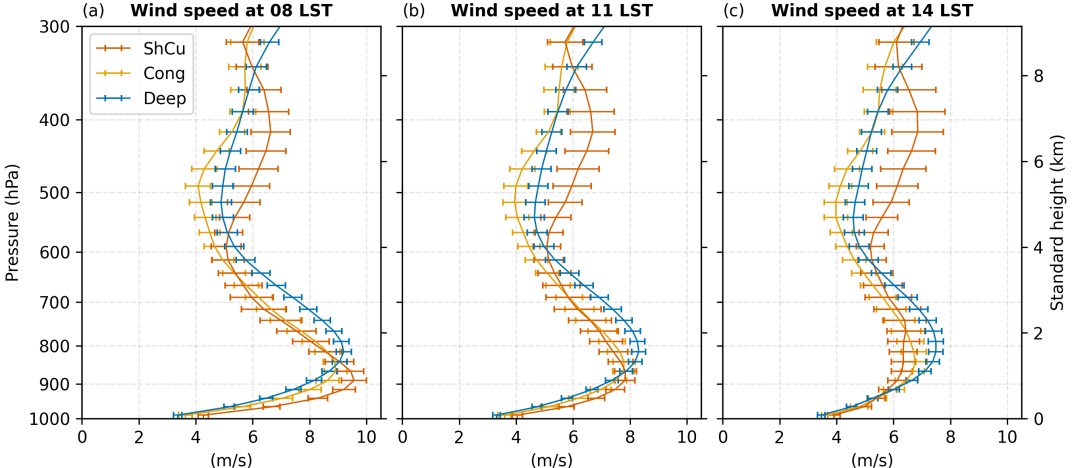

**Figure 10.** Large-scale horizontal wind speed (m/s) at (a) 08 local standard time (LST), (b) 11 LST, and (c) 14 LST for ShCu, Cong, and Deep regimes.

A hodograph at 14 LST is displayed in Fig. 11. The wind turns clockwise with increasing height: north-easterly winds
dominate in the boundary layer, approximately below 800 hPa, while easterly winds dominate in the ~700-500 hPa layer. The most notable difference in wind direction occurs in the upper troposphere, above 400 hPa. Particulary, on the Deep days, the wind stops veering, demonstrating a consistent south-easterly direction. Since the wind profiles at low- and mid-levels are comparable among the convective regimes, the veering of the wind alone only hints at a possible control mechanism for the development from the congestus to the deep phase.

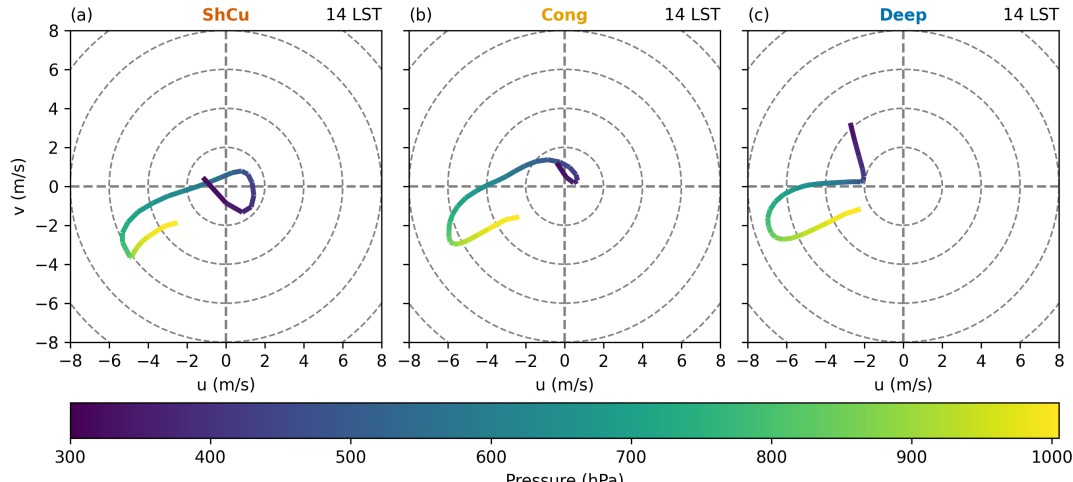

**Figure 11.** Hodograph representing the u (m/s) and v (m/s) large-scale wind components as a function of pressure at 14 LST for (a) ShCu regime, (b) Cong regime, and (c) Deep regime.

Figure 12 contains the large-scale vertical velocity ($\omega$ in hPa hr$^{-1}$) from the variational analysis at 08 LST, 11 LST, and 14 LST. In the early morning, the Deep regime shows moderate upward large-scale vertical velocity below 700 hPa, relatively greater than those associated with the ShCu and Cong regimes. Above 700 hPa, $\omega$ becomes positive (subsidence) and exhibits comparable values among the convective regimes. Around noon (11 LST), the Deep regime is dominated by upward large-scale vertical velocity below 600 hPa, while the ShCu and Cong regimes exhibit similar $\omega$ profiles with subsidence dominating

above the surface. In the afternoon, significant differences are observed between the convective regimes. The Deep regime is dominated by upward large-scale vertical velocity, especially above the 800 hPa level. The ShCu and Cong regimes exhibit important subsidence in the low levels, particularly below 700 hPa, with ShCu-subsidence assuming higher magnitudes. Tian et al. (2021) also reported a significant disparity in the large-scale vertical velocity between Deep and ShCu days, especially in the afternoon and early evening.

To evaluate the vertical wind shear, we used the bulk wind shear, which is defined as the magnitude of the vector difference of the wind at two levels. Figure 13 shows the vertical bulk wind shear for the layers 0-2 km, 0-4 km, and 0-6 km. Additionally, the bulk shear from the level of maximum wind speed below 4 km and the associated pressure level is provided in fig. A2. The 0-2 km layer exhibits a greater dependence on the diurnal cycle, with the Deep days followed by Cong days showing the most substantial wind shear at any time. Moreover, the Deep regime shows the largest difference between the maximum wind speed

below 4 km and surface wind. These results suggest that low-level vertical wind shear is related to convection development. For the 0-4 km layer, the wind shear is more similar between the convective regimes. For the 0-6 km layer, the Cong and Deep regimes exhibit similar patterns, while ShCu days are characterized by larger wind shear values and greater variability.

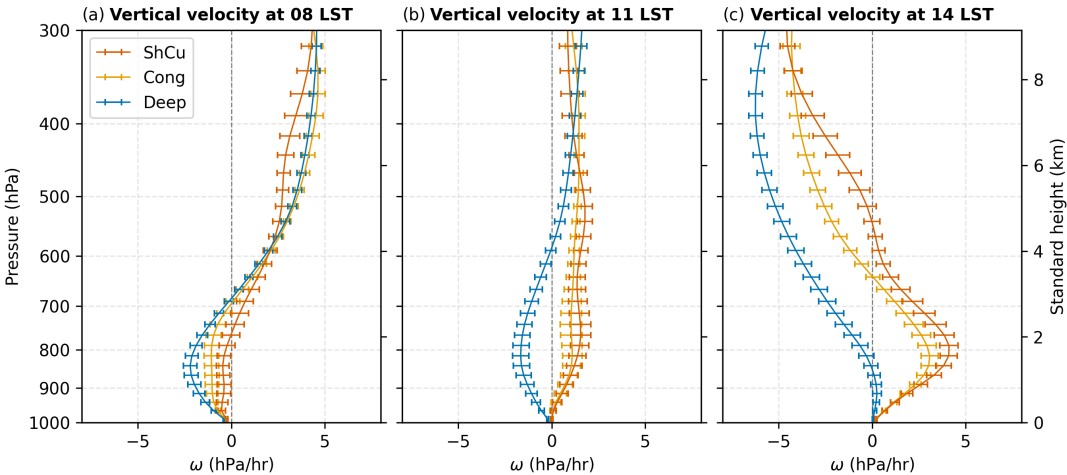

**Figure 12.** Large-scale vertical velocity ($\omega$ in hPa/hr) at (a) 08 LST, (b) 11 LST, and (c) 14 LST for ShCu, Cong, and Deep regimes.

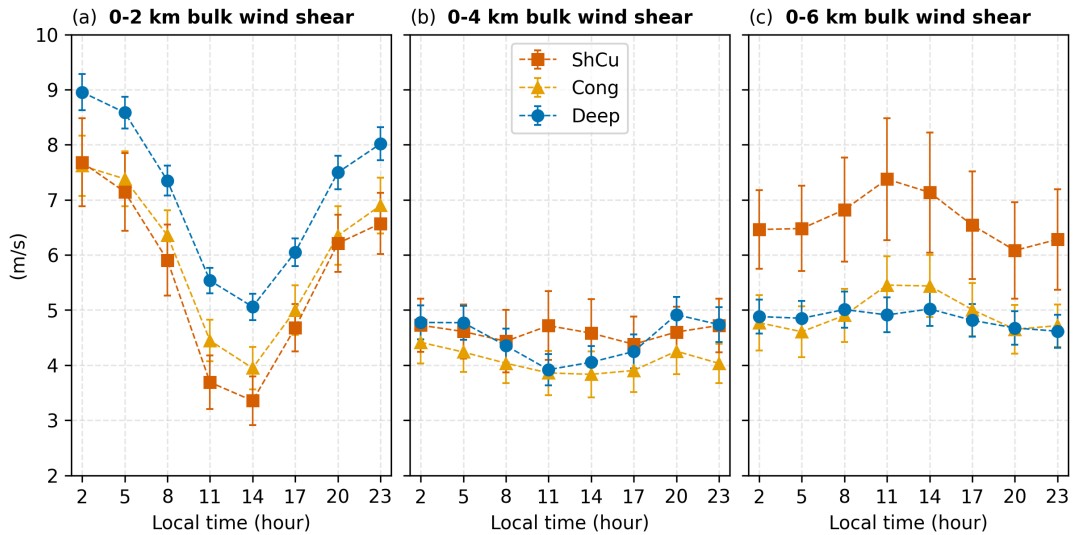

**Figure 13.** Vertical bulk wind shear (m/s) for the layers (a) 0-2 km (sfc-790 hPa), (b) 0-4 km (sfc-615 hPa), and (c) 0-6 km (sfc-465 hPa).

## 5    Conditionally Averaged Precipitation

In this section, the conditionally averaged precipitation is evaluated as a function of the main variables presumed to control the STD transition. From the previous section, we observed that convective development is primarily associated with humidity at lower levels in the early morning, while in the afternoon, it is more strongly related to upper-level moisture, instability, large-scale vertical velocity, and vertical wind shear.

Here, we explore the precipitation response to moisture using low-level and lower free troposphere CWV at 08 LST, upper-level CWV at 14 LST, MLCAPE at 14 LST, mean vertical velocity in the 1000-700 hPa and 700-300 hPa layers at 14 LST, and bulk shear magnitude for the 0-2 and 0-6 km layers at 14 LST as surrogates for vertical wind shear. Precipitation from S-band radar is averaged over the analysis domain ($100 \times 100$ km$^2$ centered at T3) and from 14 to 20 LST. The conditionally averaged precipitation corresponds to the average precipitation observed within distinct bins for each variable, which were defined as 1 mm for CWV, 2.5 J g$^{-1}$ (2500 J kg$^{-1}$) for MLCAPE, 1 hPa hr$^{-1}$ for vertical velocity, and 2 m s$^{-1}$ for bulk shear magnitude. In this analysis, it is important to highlight that we group the ShCu, Cong, and Deep regimes together and consider them collectively as local convective days.

Figure 14 consists of the conditionally averaged precipitation analysis. Large-scale afternoon vertical velocity exhibits the strongest correlation with precipitation: -0.596 in the 700-300 hPa layer and -0.570 in the 1000-700 hPa layer. Above 3 km (700-300 hPa), precipitation increases significantly from nearly zero to approximately 0.8 mm hr$^{-1}$ as vertical motion rises from nearly zero to about -9 hPa hr$^{-1}$. These correspond to the heaviest conditionally averaged precipitation rates. Interestingly, vertical wind shear, especially the low-level (0-2 km) bulk shear magnitude, exhibits the second strongest correlation with precipitation, although it has a modest value (0.348).

The lower free troposphere CWV at 08 LST demonstrates a similar correlation (0.311) with precipitation. The correlations for low-level CWV at 08 LST (0.274) and upper-level CWV at 14 LST (0.254) are also comparable. Vertical wind shear at higher levels (0-6 km layer) exhibits a weak negative correlation (-0.099) with precipitation. On the other hand, bulk shear in the 0-4 km layer (0.016) and MLCAPE (0.055) exhibit the smallest correlation values, which increase only weakly with precipitation. These results suggest that CAPE is not a good indicator of precipitation in the Amazon, which is consistent with the findings of Itterly et al. (2016); Schiro et al. (2018). Thus, days starting with excess water vapor in the lowest 3 km of the troposphere and exhibiting relatively stronger large-scale vertical velocity and low-level wind shear in the afternoon, have the highest probability of developing deep afternoon convection accompanied by heavy precipitation.

## 6 Discussion

The previous section presented the wet-season composites of ShCu, Cong, and local Deep convection days for both environmental and cloud properties measured during the GoAmazon2014/5 Campaign. The cumulonimbus clouds can extend up to 16 km of altitude, and precipitation peaks around 16-17 LST, associated with the STD transition. This is consistent with previous studies in the southern Amazon. For example, Tota et al. (2000) and Machado (2002) analyzed data from the Wet Season Atmospheric Mesoscale Campaign (WETAMC), from January to February 1999, and reported two main precipitation modes: isolated convection, which peaks in the afternoon around 16 LST, and organized convection associated with mesoscale convective systems, with a maximum during the night around 04 LST. The diurnal peak associated with deep convection is also documented more recently in Ghate and Kollias (2016) and Tian et al. (2021), who also used the GoAmazon2014/5 observations but did not focus on the wet season as in our study.

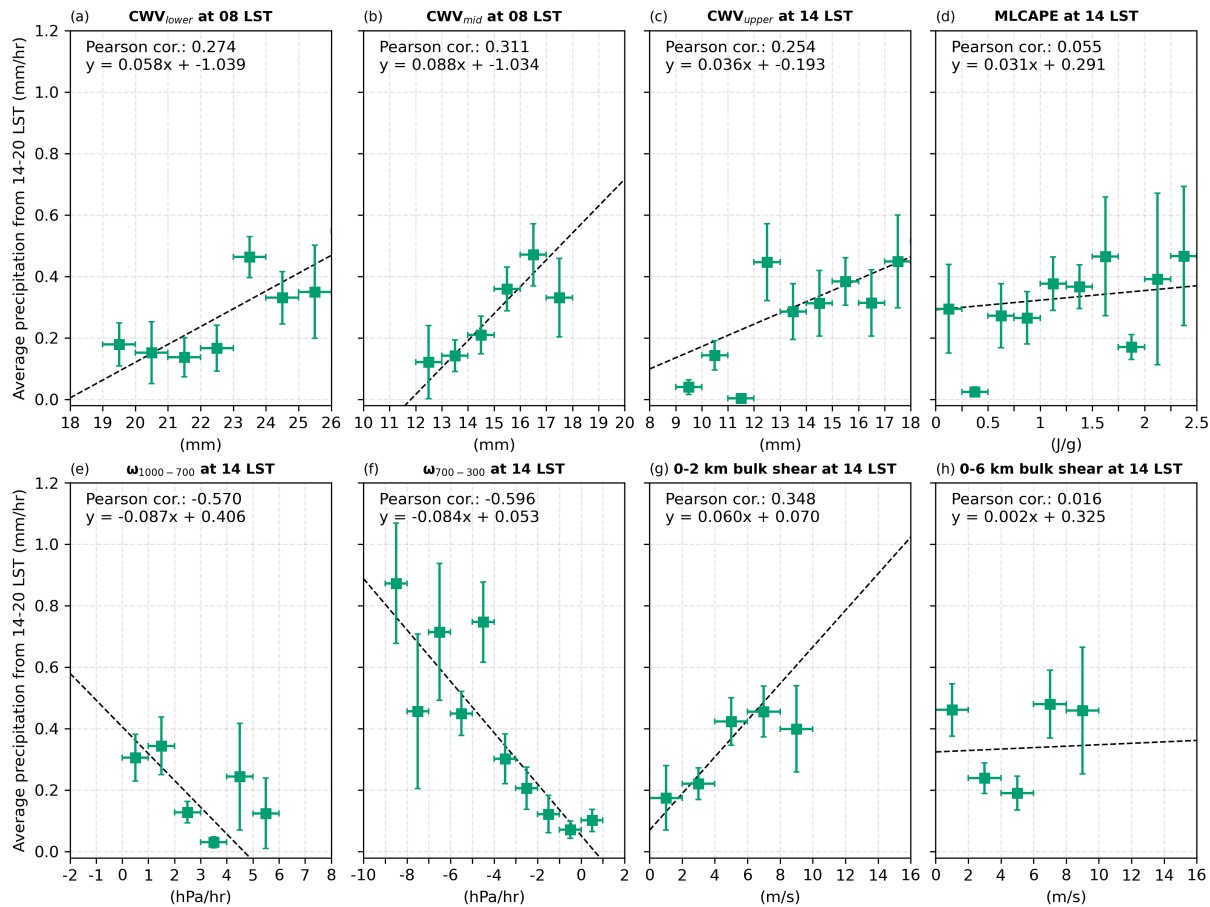

**Figure 14.** Conditionally averaged precipitation (mm/hr) to (a) low-level column water vapor (CWV, mm) at 08 LST, (b) lower free troposphere CWV at 08 LST, (c) upper levels CWV at 14 LST, (d) 100-hPa mixed-layer convective available potential energy (MLCAPE, J/g) at 14 LST, mean large-scale vertical velocity ($\omega$, hPa/hr) in the (e) 1000-700 hPa and (f) 700-300 hPa layers at 14 LST, and bulk wind shear magnitude (m/s) in the layers (g) 0-2 km and (h) 0-6 km at 14 LST. Precipitation corresponds to the mean S-band radar precipitation from 14 to 20 LST. The bins for averaging each variable are indicated by horizontal bars, which represent 1 mm for CWV, 2.5 J g$^{-1}$ (2500 J kg$^{-1}$) for MLCAPE, 1 hPa hr$^{-1}$ for vertical velocity, and 2 m s$^{-1}$ for wind shear. The conditional averaging analysis is conducted for local convective days (ShCu, Cong, and Deep regimes) indicated by green square markers. Additionally, linear least-squares regression and Pearson correlation are included for each analysis.

Our results show that deep days are associated with moister conditions in the early morning, but only in the lower troposphere, particularly below 3 km. This contrasts with the results from Itterly et al. (2016). They also noted significant differences between morning atmospheric conditions and convective intensity, but indicated that humidity in the upper troposphere also exhibits a strong relationship with convective intensity. The work by Ghate and Kollias (2016) focused on GoAmazon2014/5 observations during the dry season and observed the presence of an early morning moist layer, but elevated, between 2 and 5

365    km. In contrast, Zhuang et al. (2017) and Tian et al. (2021) found that days with deep convection exhibit increased moisture levels extending from the surface to mid-levels, regardless of the season. The discrepancies in the anomaly of the early morning moisture layer with convective intensity among these studies are likely attributed to the data and procedures used to identify convective days. Only Ghate and Kollias (2016); Zhuang et al. (2017); Tian et al. (2021) specifically utilized the GoAmazon2014/5 observations. While Ghate and Kollias (2016) differentiated between days with no precipitation (cumulus days) and

those with surface precipitation rates exceeding 0.05 mm h$^{-1}$ (precipitating days), both Zhuang et al. (2017) and Tian et al. (2021) relied on cloud boundaries, albeit with different definitions for identifying convective cloud regimes. Furthermore, all these studies relied on visual inspection to identify and eliminate cases linked to MCSs. Here, we have specifically developed a consistent systematic criterion (item v, section 2.3) based on the contiguous area of the S-band radar precipitation to identify these days and remove them from our analysis. These differences in methods contribute to differences in the selection

of convective days. Particularly in Ghate and Kollias (2016), the identification of convective days based on a single surface precipitation threshold might not be distinguishing the varying levels of convective intensity observed in the Amazon. Additionally, relying solely on site measurements ignores the surrounding areas, which could be important for gaining a broader perspective on the convective intensity of a given day. This could explain why only Ghate and Kollias (2016) observed a moist layer extending from 2 km rather than from the surface.

Based on the water vapor convergence derived from the surface water balance, we observed that during ShCu and Cong days, there is a predominant water vapor divergence. In contrast, the Deep regime days are neutral until early afternoon and show a significant water vapor convergence between 14 and 18 LST, coinciding with the STD transition. This aligns well with the convergence timescale preceding the STD transition reported by Adams et al. (2013), based solely on CWV observations. More recently, Zhuang et al. (2017) assessed the accumulated water for the convergence term through diagnostic data derived from

ECMWF model runs designed for the GoAmazon2014/5 experiment. During the wet season, they found neutral divergence during shallow days, while water vapor roughly increased linearly throughout the deep convective days. Although Zhuang et al. (2017) and our findings indicate a dominance of water vapor convergence on Deep days, the behavior differs significantly in our analysis, as we observed a relevant divergence during the daytime of ShCu days, and the convergence dominates Deep days only in the afternoon. These differences suggest that the ECMWF model runs may not have accurately captured the

diurnal cycle of convection, misrepresenting advection and convergence. This is a well-known problem of models that rely on convective parameterizations (Betts, 2002; Betts and Jakob, 2002; Grabowski et al., 2006), and that is why the variational analysis dataset assimilated the S-Band radar precipitation and surface fluxes from GoAmazon2014/5 experiment (Tang et al., 2016).

        We observed that mixed-layer CAPE is lower on ShCu days and higher on Deep days, with the difference increasing from

nighttime (2 LST) to early afternoon (14 LST). However, when the conditionally averaged precipitation is considered, we found that MLCAPE is only weakly correlated with afternoon precipitation. Itterly et al. (2016) also found that CAPE is an unreliable indicator of convective activity in the Amazon. Moreover, Schiro et al. (2018) investigated the relationship between buoyancy and precipitation and showed that CAPE is weakly related to precipitation when deep-inflow mixing is not included that represents the entrainment of dry air in the parcel.

For the large-scale wind, we observed that precipitation increases significantly with afternoon upward vertical velocity, especially above 3 km, and moderately with increasing low-level wind shear (0-2 km). At upper levels, vertical wind shear has a negative impact on convection, although its correlation with precipitation is relatively weak compared to low-level shear. Using GoAmazon2014/5 observations, Zhuang et al. (2017) reported a similar pattern of bulk wind shear during the wet season. They suggested that strong wind shear would favor convection. However, during the wet and transition seasons, Deep days are associated with weaker upper-level shear. Thus, they suggested that wind shear may have no impact or could even hinder convection. Here, our results support that strong wind shear at lower levels favors convection, while weaker shear at upper levels plays a minor role in convection. Therefore, low-level wind shear may facilitate the development of convection during the wet season. On the other hand, Chakraborty et al. (2018) suggested that shallow convection is associated with stronger low-level and weaker upper-level shear intensity during the transition season, which is the opposite of what was shown by Zhuang et al. (2017) and observed in our study focused on the wet season. This difference could be attributed to variations in the data employed, procedures for identifying convective clouds, and potentially because Chakraborty et al. (2018) did not evaluate the diurnal cycle. Instead, they considered radiosonde wind measurements within 2 hours before the development of shallow convection, in scenarios where it did or did not evolve into deep convection. On the other hand, Zhuang et al. (2017) used radar wind profiler data to specifically assess low-level wind shear. Moreover, our findings generally agree with Tian et al. (2021) that mid-tropospheric vertical wind shear could significantly suppress the vertical development of convection, although to a lesser extent compared to low-level wind shear.

## 7   Conclusions

We analyzed measurements from the GoAmazon2014/5 field campaign in the central Amazon with the goal of assessing possible controlling mechanisms of the shallow-to-deep convective transition. We classified wet season days into shallow (ShCu), congestus (Cong), and Deep regimes, purposely excluding mesoscale systems to focus on locally-driven deep convection. Unlike previous studies, we used an objective and reproducible method to identify and exclude days dominated by organized convection. Additionally, the bootstrap resampling method was employed in analyzing the environmental controls of each convective regime, permitting a robust assessment of the statistical significance of the results despite the relatively small sample size. The Deep regime is characterized by early morning moister conditions extending from the surface to the lower free troposphere. This agrees with some previous studies (Zhuang et al., 2017; Tian et al., 2021), but differs from others (Ghate and Kollias, 2016). We also found that ShCu and Cong days are characterized by water vapor divergence, while Deep days start neutrally (from 8 to 13 LST) and develop strong water vapor convergence in the afternoon (from 13 to 17 LST), which peaks coincide with the STD transition (Adams et al., 2013). This is the first analysis of the moisture convergence using GoAmazon2014/5 observations, as previous studies relied on models that use parameterized convection (Zhuang et al., 2017).

In contrast to several previous studies that emphasized the role of humidity in convective activity in the Amazon region (Itterly et al., 2016; Ghate and Kollias, 2016; Schiro et al., 2016), our results indicate that precipitation may exhibit an even stronger correlation with large-scale vertical velocity than with the moisture content in any level or at any time of the day.

Nevertheless, it should be noted that vertical velocity is challenging to assess through observations. Furthermore, we have found that precipitation increases with low-level wind shear, displaying a correlation comparable to that observed for moisture content. This correlation has not been evaluated by previous works, which even diverged in their interpretation of the wind shear's role on the STD transition (e.g., Zhuang et al., 2017 and Chakraborty et al., 2018).

Finally, our results suggest that pre-convective humidity in the lower troposphere and diurnal large-scale vertical velocity, water vapor convergence, and low-level wind shear represent the primary environmental factors influencing convective intensity. This indicates that dynamic factors may play a more prominent role in convection during the Amazon wet season than during others. To further disentangle the roles of these environmental controls, we recommend conducting numerical experiments using cloud-resolving models that could, for instance, examine the sensitivity of the STD transition to changes in moisture or wind shear at different atmospheric levels. While numerous studies have explored these recent observations over the Amazon, only a few have utilized high-resolution simulations to investigate the environmental controls of convection (e.g., Cecchini et al., 2022). The VARANAL large-scale data could be used to force cloud-resolving models or even directly evaluate the water budget (but note that it operates on a different spatial and temporal scale than the one analyzed in this study). Likewise, longer-term, high-density observational networks in the Amazon, such as Adams et al. (2015), would be of great value for constraining or evaluating numerical model results.

*Data availability.* The observational data used in this research are publicly available at https://www.arm.gov/research/campaigns/amf2014goamazon. The variational analysis large-scale forcing data for the GoAmazon2014/5 experiment (Tang et al., 2016) is available at the ARM Archive: http://iop.archive.arm.gov/arm-iop/0eval-data/xie/scm-forcing/iop_at_mao/.

## Appendix A: Supplemental Material

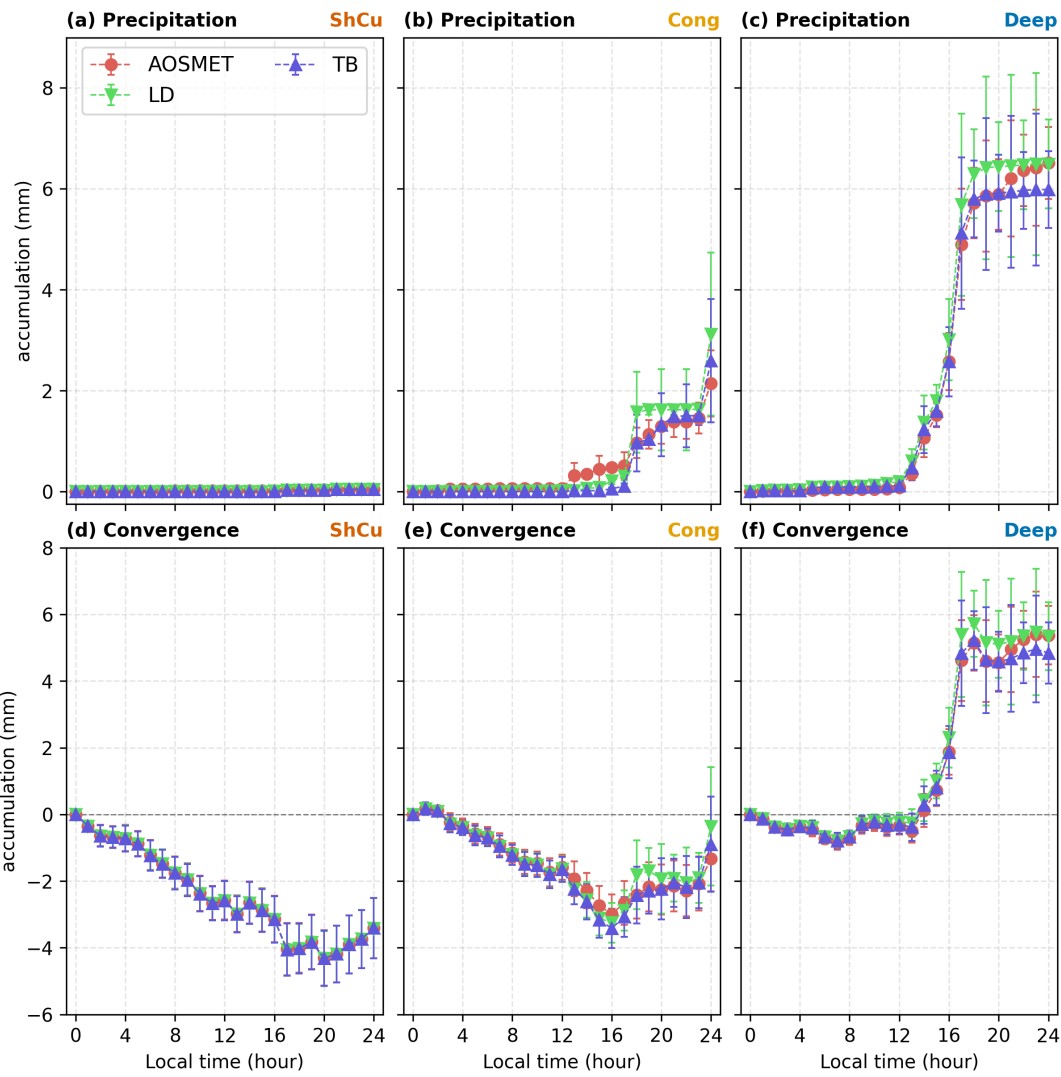

**Figure A1.** Accumulated surface precipitation (mm) for the aerosol observing system surface data, tipping bucket, and laser disdrometer are shown in the upper panels (a-c). The corresponding convergence (mm) term for each instrument is displayed in the lower panels (d-f). The ShCu, Cong, and Deep regime results are shown in the first, second and third columns, respectively.

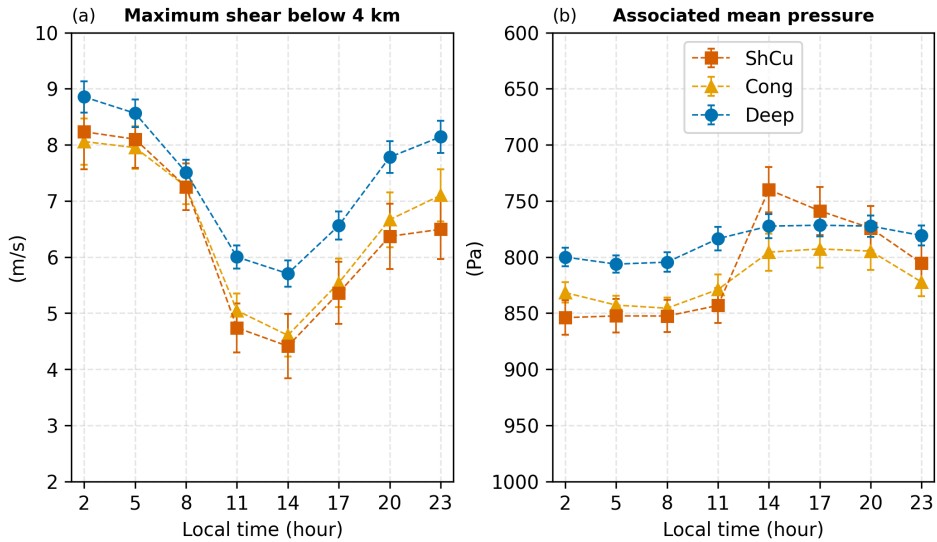

**Figure A2.** (a) Vertical bulk wind shear from the level of maximum wind speed below 4 km and (b) the associated pressure level.

**Table A1.** Data

| Name | Description |
|---|---|
| Cloud mask (Giangrande et al., 2017) | Combines multiple sources of cloud data to provide a high-resolution temporal (30 sec) vertical profile of cloud type, including shallow, congestus, and deep. Cloud frequency profile is also calculated as the cloud counting fraction over 12 minutes. |
| S-band radar (Schumacher and Funk, 2018a, b) | Volume data of reflectivity and derived precipitation rate at 2 km. Rain cover is calculated as the fraction of reflectivity $> 20$ dBZ over a 100x100 $km^2$ analysis domain. These data have a temporal resolution of 12 minutes, a horizontal resolution of 2 km, and a vertical resolution of 500 m. |
| Surface precipitation | We employ three source of surface precipitation data, namely, aerosol observing system surface data, tipping bucket, and laser disdrometer. They have only a temporal dimension, for which we use an average over 12 minutes or 1 hour. |
| Atmospheric State | Radiosonde data has a latency of 6 hours (launches at 02, 08, 14, and 20 LST). We utilized sounding profiles covering at least 8 km of the atmosphere. A derived planetary boundary layer height is also used. Additional variables, such as CWV, MLCAPE, and MLCIN are calculated. |
| Surface fluxes | Hourly average data from the ARM best-estimate dataset based on observations from the Eddy Correlation Flux Measurement. Surface evaporation is calculated and employed in the water budget analysis. |
| Water content | Hourly average data of CWV and LWP taken from the microwave radiometer are employed in the water budget analysis. |
| Large-scale wind field | Based on the variational analysis. It corresponds to a 3-hour average and a domain average of $\sim$100 km. |

*Author contributions.* Leandro Viscardi: Investigation, Methodology, Data curation, Formal analysis, Visualization, Writing - Original draft preparation. Giuseppe Torri: Investigation, Validation, Resources, Writing - review & editing. David Adams: Investigation, Writing - review & editing. Henrique Barbosa: Conceptualization, Investigation, Resources, Supervision, Writing - review & editing.

*Competing interests.* The authors declare that they have no conflict of interest.

*Acknowledgements.* We acknowledge the data from the Atmospheric Radiation Measurement (ARM) Program sponsored by the U.S. Department of Energy, Office of Science, Office of Biological and Environmental Research, Climate and Environmental Sciences Division. L.A.M.V. acknowledges the Brazilian National Council for Scientific and Technological Development (CNPq) graduate fellowship (grant

number 148652/2019-0) for supporting his doctoral studies at the University of São Paulo, Brazil. Additionally, gratitude is extended to the Coordination for the Improvement of Higher Education Personnel (CAPES) fellowship (grant number 88887.571091/2020-00) for supporting 6 months of a visiting graduate student program at the University of Hawaiʻi at Mānoa, United States, during his doctoral journey. H.M.J.B acknowledges support by the U.S. Department of Energy, Office of Science, Biological and Environmental Research program under Award Number DE-SC-0023058.

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
