# Peer review of "Environmental Controls on Isolated Convection during the Amazonian Wet Season"

_EGUsphere, 2023_

## Referee Comment (RC2)

**Review of Viscardi et al. 2023, ACP**
**"Environmental Controls on Isolated Convection during the Amazonian Wet Season"**

This study aims to characterize the environmental factors associated with the predominance of different convection modes over the Amazon basin during the wet season. Based on a systematic classification of the convection events predominant from late morning to evening in a $100 \times 100$ km sub-domain located in the Central Amazon, the authors obtain mean composites of 1D and 2D measurements and compare their most relevant patterns among different convection regimes.

Overall, the paper is well structured, and results are put into context by comparing them to the preceding literature. Although some of the analyses resemble previously published results, the use of different data and methodology compared to those adds relevance to this manuscript, especially considering the difficulty of generalizing in-situ atmospheric observations. Moreover, the moisture convergence analysis has not been previously explored using this set of observations, to the best of my knowledge.

However, I have some concerns regarding the impact of measurement uncertainties on the results, as well as the statistical significance of the findings. Additionally, I would like to see some aspects of the methodology clarified. I also found that some statements need further justification or consideration. Therefore, I recommend major revisions.

**Major comments**

1. One of my main concerns regarding the methodology employed in this study, is the impact of the errors in the retrieval of the water budget terms, on the calculation of the divergence term. I support the ideas suggested by the previous reviewer to assess this impact, such as repeating the analysis with a different data source, and further discussing the shortcomings and implication of these issues via the inclusion of a residual error term in the budget equation.

   Relatedly, the control volume over which the budget analysis is performed needs to be defined. Please discuss how well do the measured profiles represent the control volume, and its implications for the analysis.

   In the water budget equation, you have used the hydrostatic approximation $-\rho.\partial z = \partial p/g$, whereas it is employed to analyze cases where the non-hydrostatic component may be high, especially in the deep convection cases. The expression employed to calculate the first term may thus induce errors in CVV and LWP that ultimately affect the calculation of the divergence term, for a given E and P. Please discuss the implications of this assumption for the results and justify the validity of your approach.

2. The sizes of the regime samples seem relatively low, therefore I suggest analyzing the statistical significance of the differences found between the regimes for the different aspects considered in the results section.

3. For a reader that is not familiar with the data employed here, it may result difficult to keep track of the different measurement sources throughout the different sections, therefore I suggest including a table in the methodology section with the name and short description of each data source, along with the dimensionality of the data and its use in the paper.

4. Is the 2% threshold of 'rain coverage' used in the convection regime classification arbitrary? Please discuss how would the choice of a different threshold affect the results? I suggest showing

histograms of the rain coverage at each height interval ($h < 3$km, $3$km$< h < 8$km, and $h > 8$km), for each cloud regime. This would complement Fig 5, and help understanding how much would the choice of the threshold impact the classification.

5. The adequateness of the 'bulk wind shear' (defined here as the difference in wind speed between two specific levels) to analyze the impact of wind shear on convection development is questionable. Different from other magnitudes that represent the entire atmospheric column or portions of its depth, the so-called 'bulk wind shear' depends on punctual wind measurements along the atmospheric column, neglecting the behavior at other levels. For example, as mentioned in section 4.4, the difference in the wind speed from 0 to 2km is generally higher in the deep-convection compared to the shallow-convection days, thus the authors conclude that 'low-level vertical wind shear is related to convection development'. However, Fig. 10a shows that the vertical wind shear is actually more intense in shallow cases at 8h, when considering the levels of maximum LLJ and the surface as a reference. A more apples-to-apples comparison could be done by, for example, calculating the bulk shear from the level of maximum wind to the surface, and complementing this information with the height of such maximum.

**Minor comments**

- **line 14:** Missing comma after 'even moisture'

- **line 46:** Define 'mid-levels', for clarity

- **line 72:** Typing error: 'a city border the Rio Negro'

- **line 77-78:** Suggested: '... a cloud mask based on time-height profiles of the cloud location'

- **line 83:** What do you mean by 'cloud frequency profile'? Relatedly, in Fig 5's caption, what is the percentage cloud frequency relative to? Is it the total number of cases? Please clarify.

- **line 107:** 'water vapor mixing ratio'

- **line 112:** Do you mean 'the 100-hPa depth layer immediately above the surface'?

- **line 126:** Need to define the control volume over which the conservation equation is being applied.

- **line 129:** Please rephrase for clarity: 'E and P correspond to the water mass fluxes associated with surface evaporation and precipitation' (specifying 'surface evaporation' avoids confusing this with internal phase changes in the control volume)

- **line 131:** 'with the divergence of water vapor'

- **line 132:** I believe this statement is incorrect. Since the variable here is total water mass mixing ratio, it is not affected by phase transitions associated with cloud formation in the control volume.

- **line 134:** I believe there is a mistake here. The terms in Equation 1 have units of mass flux, such as kg/m$^2$/s. Dividing by the density of liquid water would give units of m/s, which does not seem to be the intention. Moreover, to express the equation in terms of liquid and vapor water paths, it is not necessary to divide by anything, since the integral of the total water mixing ratio divided by the gravity acceleration with respect to pressure already represents the total water path in a hydrostatic atmosphere. This total water path can then be separated into vapor, liquid, and ice terms, as you mention below.

- **line 134:** 'ignore' should be 'neglect'

- **line 134:** Are you neglecting the time variation of the ice mass or the presence of ice itself in the analysis? Does the observed LWP you use include all condensate or does it discriminate between ice and liquid water?

- **line 138:** If you follow the suggestion I mentioned earlier and avoid dividing by any additional factors, the terms in the equation can be represented as EVAP = E and PREC = P. I recommend maintaining one naming convention for consistency and clarity.

- **line 141:** The data in Fig 9 seems to have a higher frequency, intervals seem to be of 12 min.

- **line 152:** Do these times define what you call 'diurnal cycle' throughout the paper? If so, please clearly define the term 'diurnal cycle' when it is first used in the text, to ensure that readers understand the time period being discussed.

- **line 153:** By 'rain coverage' are you referring to large hydrometeors, independent of the phase. Please clarify.

- **line 157-164:** By 'precipitation', do you mean 'rain coverage'? If so, please correct the text here for consistency.

- **line 165-166:** Above, you list the conditions for each type of convection regime, while here you list conditions that apply to all regimes, right? If so, I suggest clarifying this.

- **line 173:** I suggest 'the propagating-convection category occurs more frequently during the wet season', for clarity.

- **line 195:** I suggest rewording 'because' to: 'associated with'.

- **line 200:** Suggested rewording: 'coincides' to 'is consistent with'

- **line 207:** Please add the figure numbers.

- **Fig 6c,d:** Is the difference between the profiles for different regimes larger than the standard deviation of the measurements? The length of the bars is not evident in panels a and b, so at least commenting on that would be helpful.

- **line 214:** I suggest providing a quantitative measure instead of 'is quite similar'. Maybe something like: 'the difference between the moisture profiles for different regimes is less than 1/x of the maximum difference at lower levels'.

- **line 217:** Suddenly starting to talk about the tropical oceans reads a bit strange. If the goal is to mention contrasting results, I suggest rephrasing it to smooth the transition, for example: 'Moreover, these results contrast with studies over tropical oceans, where free-tropospheric humidity has been shown to play a more significant role...'.

- **line 223:** Where it says '...at 2 LST', it should be noted that since the classification was done for times between 10-20 LST, measurements at other times of the day might include different types of cloud than those in the regime in question.

- **line 231:** Where is says 'despite slightly larger latent heat', add a reference to the section of the manuscript where this is shown.

- **line 233:** 'water vapor convergence', ditto

- **line 242:** Please maintain the naming convention (MLCAPE for CAPE).

- **line 249:** It appears that you are calculating 'accumulated values' by numerically integrating the mm/day values over time intervals of approximately 12 minutes. However, it's unclear how the resulting values are still being expressed in mm. Given that the time step (dt) is 1/120 days, it seems that the resulting unit should be mm multiplied by the time step, which would be mm/120.

- **line 249:** Instead of 'variation in LWC', I suggest using '$\partial_t$LWC', for clarify.

- **line 250:** 'dominated' seems to imply that CONV is the largest term among those in the budget analysis, so I would suggest rewording, for example, 'shows mostly water divergence'.

- **line 251:** The 'divergence' term responds to changes in the total water, not just water vapor. I understand that the convergence is expected to occur in the form of water vapor, since horizontal transport of condensate is less likely, but this should be somehow clarified, at least in a footnote.

- **line 251:** Suggest: 'relatively neutral'

- **line 256:** I suggest framing the discussion slightly different: instead of saying that high evaporation balancing the divergence leads to low precipitation, it is seems more intuitive to think it as high evaporation and low precipitation requiring strong divergence for closure.

- **line 257:** Ditto

- **line 258:** Is CWP a typo? Do you mean CWV? As commented before, it is clearer to say $\partial_t CWV$ than 'the variation in CWV'.

- **line 259:** What do you mean by 'accumulation of water vapor'? Are you referring to Fig. 9d? If so, please clarify. Similarly, clarify the panels you are referring to in the rest of the text.

- **line 269:** Please expand briefly on how the LLJ is affected by the PBL growth. Do you mean that the PBL growth slows down the LLJ? Please clarify that this is one possible explanation, since other factors may be involved in regulating the intensity of the LLJ.

- **line 271:** Better refer to the layer between $\sim 600$ and $\sim 350$ hPa or so, since at 300 hPa the wind tends to be intense in the deep case.

- **line 275:** Suggested: 'The most notable difference'

- **Fig 9, caption:** Instead of referring to the number of the subsection, I suggest referring to the number of the equation.

- **line 295-297:** Although the 6-km level is taken as a reference here, the corresponding 'bump' in the wind profile starts at 4km and reaches 8km height (as you mentioned in the discussion of Fig. 10). Moreover, it would be incorrect to say that only because it happens at levels higher than the top of the cloud, the wind pattern does not matter for the convection development. For example, the wind profile impacts the gravity wave pattern induced by convection itself, and these waves may then feedback on the dynamics of the convective clouds [e.g. CHK86].

- **line 308:** 2.5 J/g = 2500 J/Kg, not 250 J/kg. But there is no need to mention both values anyway.

- **line 321:** Based on this results, you can only state that CAPE is not a good indicator for precipitation. This analysis doesn't involve convection strength, but precipitation only.

- **Fig 14, caption:** 'average' should be 'averaged'. Also, what do the vertical bars mean?

- **line 328:** 'Results showed that isolated deep convection is associated with more extensive and longer-lived clouds throughout the diurnal cycle.' Do you mean compared to shallower clouds? I suggest omitting this statement, as such feature can be considered 'common knowledge'.

- **line 339:** Please clarify what you mean by 'noted significant differences between morning atmospheric conditions and convective intensity'.

- **line 366:** It should be 'captured', instead of 'capture'.

- **line 380-382:** Did they suggest that wind shear would both favor and hinder convection? Please clarify.

- **line 406:** 'this value', which one?

**References**

[CHK86] Terry L. Clark, Thomas Hauf, and Joachim P. Kuettner. Convectively forced internal gravity waves: Results from two-dimensional numerical experiments. *Quarterly Journal of the Royal Meteorological Society*, 112(474):899–925, 1986.

---

## Author Comment (AC1)

**General Response to Referee #1**

Original referee comments are in blue.
Our responses are in black with regular font format. Text from the updated manuscript:

*Appears in italics with 1⁄2 inch indentation with the removed parts exhibited with a  and red color while new text added is shown in green.*

**Summary of RC1 comments:**

The authors introduced a clear and robust regime classification methodology. Their novel findings underscore the significance of low-level wind shear and estimated moisture flux convergence for each regime.

We are grateful for the encouraging comments and would like to acknowledge the thoughtful revision, which helped improve our manuscript.

However, the authors' broad investigation of potential forcings may have compromised the depth of certain analyses, such as the water budget closure, which merits further exploration. (1) Additionally, they should address uncertainties related to rain gauge-measured precipitation and surface fluxes.

(2) Suggestions include incorporating symbolic error terms to account for measurement errors and exploring error budget analyses within each regime.

(3) Furthermore, they should consider alternative precipitation sources for water budget closure could strengthen the study.

In order to provide uncertainty estimates with respect to the water budget, we followed the reviewers' suggestions and considered different sources of precipitation in our updated analysis. In addition to the surface AOSMET precipitation [1], we are now also using the tipping bucket [2] and laser disdrometer (LD) [3]. Figure 9 in the current manuscript will be replaced by the figure below, which displays the updated results.

[Figure]

**Figure 9.** Surface water balance for (a,d) ShCu regime, (b,e) Cong regime, and (c,f) Deep regime. The upper panels represent the rates of change (mm/day) for water vapor convergence (CONV), evaporation (EVAP), precipitation (PREC), and the time derivatives of column water vapor ($\partial_t CWV$, where $\partial_t = \partial/\partial_t$) and liquid water path ($\partial_t LWP$). The lower panels display the accumulated water amounts for each term in the water budget along the day (mm). Note that CWV and LWP changes rely on microwave radiometer observations, evaporation on eddy correlation flux measurements, and precipitation  is estimated utilizing different sources, namely, the aerosol observing system surface data, tipping bucket, and laser disdrometer. The water vapor convergence term is estimated as a residual in the water budget equation (Section 2.2).

For the rate amounts, the time derivative of CWV or LWP corresponds to the microwave radiometer (MWR) data source. Evaporation rate is calculated from the Eddy Correlation (ECOR) Flux Measurement. Precipitation is the mean composite from the AOSMET, tipping bucket, and laser disdrometer sources. The precipitation uncertainty is estimated from the standard deviation of the different sources. Mean convergence is estimated from the water balance closure, i.e., equation (2) at line 137 from the manuscript. The standard deviation of mean convergence is estimated from the standard deviation of the mean $\partial_t CWV$, $\partial_t LWP$, EVAP, and PREC, i.e.,

$$\sigma_m = \sqrt{\sigma_m(\partial_t CWV)^2 + \sigma_m(\partial_t LWP)^2 + \sigma_m(EVAP)^2 + \sigma_m(PREC)^2} \quad .$$

Mean accumulation amounts and their standard deviations are obtained analogously. Note that the convergence uncertainty is mainly attributed to the uncertainty of CWV. More importantly, the main message of our analysis remains unchanged despite the uncertainties included, i,e., shallow days are dominated by divergence while deep days by convergence. Finally, to provide additional information to the readers, we will also include the precipitation and convergence term associated with each data source in the Supplementary Materials, as indicated by the Figure below:

[Figure]

**Figure A1.** Accumulated surface precipitation (mm) for the aerosol observing system surface data, tipping bucket, and laser disdrometer are shown in the upper panels (a-c). The corresponding convergence (mm) term for each instrument is displayed in the lower panels (d-f). The ShCu, Cong, and Deep regime results are shown in the first, second and third columns, respectively.

With this improved analysis, we believe that all the uncertainties related to the water budget are resolved consistently, given the available observational data. Likewise, the results are statistically significant.

To include the updated version of the surface water balance, the current manuscript must be modified as follows:

**Line 115**:

*Instead of using the S-band radar precipitation, the water balance analysis  utilizes a combination of rain-gauge source measurements to provide a more robust estimation of the mean surface precipitation and its uncertainties. Specifically, we use the surface AOSMET precipitation [1], tipping bucket [2], and a laser disdrometer (LD) [3].*

**Line 141-143**:

Specifically, CWV and LWP are based on the MWR. ECOR latent heat flux data is utilized for estimating the evaporation term. Precipitation is  obtained utilizing different sources, namely, aerosol observing system data, tipping bucket, and laser disdrometer (Section 2.1).

**Line 248-250:**

The water balance results are shown in Fig. 9, with the top panels showing the hourly average rate values (mm/day) and the bottom panels showing the corresponding accumulated values (mm). The separate precipitation and convergence associated with each rain gauge used to estimate the mean surface precipitation and its uncertainty in the water budget results are provided in fig. A1. First, we notice that the variation in LWP appears negligible, and it does not contribute significantly to the water budget.

**Line 254-255:**

At day's end, the ShCu regime is estimated to lose 3.4 mm of water vapor due to divergence, while the Cong regime loses  0.9 mm. In contrast, the Deep regime gains  5.2 mm of water vapor due to convergence.

**Figure 9, page 14:**

**Figure 9 is replaced by the updated one, shown above. To consider the new sources of surface precipitation, the caption is updated as follows:**

Note that CWV and LWP changes rely on microwave radiometer observations, evaporation on eddy correlation flux measurements, and precipitation  is estimated utilizing different sources, namely, the aerosol observing system surface data, tipping bucket, and laser disdrometer.

(4) Enhancing visualization clarity in Figure 6.

We updated Fig. 6 to limit the y-axis from surface to 5 km and also included the estimated standard deviation for the difference in potential temperature and mixing ratio $(\sqrt{\sigma_m(Deep)^2 + \sigma_m(ShCu\ or\ Cong)^2}\ )$, as shown on figure below.

[Figure]

**Figure 6.** Atmospheric conditions. (a) Potential temperature (K) and (b) water vapor mixing ratio (g/kg) at 08 LST radiosonde observations. The corresponding convective regime differences (Deep-ShCu and Deep-Cong) for potential temperature ($\Delta\theta$, c) and mixing ratio ($\Delta r_v$, d) are also included.

Despite the uncertainties in the difference of mixing ratio between Deep and ShCu composites, values are greater than 0 from the surface up to 3 km. Hence, our main message remains the same: Deep days are relatively moister than shallow days only in the lower levels. The differences between congestus and deep days are less pronounced, albeit the deep composite tends to be slightly moister than the congestus profile.

(5) Suggestions for future research, such as applying constrained variational analysis for moisture flux convergence estimation, are recommended.

We thank the reviewer for the suggestion. We agree that the variation analysis could be an interesting source of data to investigate the water balance, particularly because all the inputs and outputs of the model estimates are forced to be physically consistent.

However, the VARANAL horizontal scale is ~100 km and the temporal scale is about 3 hours, which is not exactly the same scale involved in our manuscript. Thus, further analysis would be necessary to directly compare the constrained analysis and our

results, if we were to include here. That is why we intentionally focused solely on direct observations in this current manuscript.

Nonetheless, we added the following in the conclusions of the manuscript to indicate that future studies should look into VARANAL (or Data Assimilation in general) to investigate further the water budget of the STD transition:

Lines 415-418:
While numerous studies have explored these recent observations over the Amazon, only a few have utilized high-resolution simulations to investigate the environmental controls of convection (e.g., Cecchini et al., 2022). The VARANAL large-scale data could be used to force cloud-resolving models or even directly evaluate the water budget (but note that it operates on a different spatial and temporal scale than the one analyzed in this study). Likewise, longer-term, high-density observational networks in the Amazon, such as Adams et al. (2015), would be of great value for constraining or evaluating numerical model results.

(6) The authors should consider presenting conditionally average precipitation findings related to rain gauge-averaged sources between 14-20 LST as an appendix or mention if differences are negligible.

We considered it, but this would cause a mismatch of spatial scales. We remind the reviewer that our "conditional analysis" uses the S-band radar precipitation data averaged over a domain of 100x100 km$^2$. Moreover, the precipitation in Figure 14 is the afternoon precipitation (14-20 LST), while the variables are shown either at 8 or 14 LST. Last but not least, this analysis includes all days; hence it is different from all the previous figures that distinguished between ShCu, Cong, and Deep days.

The other sources of precipitation, such as AOSMET, TB, and LD, are local point measurements. Thus, unless the rain is exactly over the T3 site, only the S-band radar will capture it. This significantly reduces the correlation between the precipitation metric (which is now local) and the evaluated variables (atmospheric state based on sounding measurements or large-scale wind). To support our argument, the figure below shows the conditional analysis based on the AOSMET precipitation, and no correlation is seen.

Essentially, our methods evaluate how the atmospheric state in the morning or a few hours preceding the STD convective transition (at 14 LST) relates to the mean precipitation over a large-scale domain during the entire afternoon. That is why we employed the S-band radar for this particular analysis.

To include local sources of precipitation, a possible alternative approach would be to follow the methods in [4], i.e., apply a more instantaneous comparison (average precipitation and other variables at the same time range) instead of evaluating 8 or 14 LST atmospheric state conditionally to the diurnal (14-20 LST) averaged precipitation. However, this would be more similar to evaluations conducted on previous studies (except that they did not evaluate large-scale wind).

[Figure]

(7) Additionally, there have been a few technical corrections reported regarding the text, aimed at enhancing sentence structures.

- Lines 222-223: CWV difference at 8 LST doesn't appear statistically significant for Deep – Cong
**Answer:** Thank you for your observation. The text was modified as follows:

Previous version:

Updated version:

The differences in CWV among Cong and Deep regimes show little statistical significance from nighttime to early morning, while their difference is maximum at 14 LST when it reaches 2.1 mm.

- Lines 226-227: Change "A similar, albeit less pronounced, pattern is also observed in the lower free troposphere."

  To e.g.,:

  "A similar, albeit less pronounced diurnal moistening is also observed in CWVmid for all regimes."
  **Answer:** done

- Lines 230-231: Remove ", and associated with low-level moisture divergence, which would explain the slower accumulation of moisture below 700 hPa despite slightly larger latent heat." Current logic is too speculative.
  **Answer:** done

- Lines 239-242: Avoid comparing Cong vs. Deep when differences are negligible. Consider reframing certain comparisons as Cong+Deep vs. ShCu in this section in general e.g.,

  "In the afternoon, MLCAPE is higher for the Deep regime (1237 J kg−1), a few hours before the late afternoon STD transition, slightly surpassing the value for the Cong regime (1111 J kg−1) and significantly exceeding the value for the ShCu regime (671 J kg−1)."

  could be changed to:

  "At 14 LST, MLCAPE for the Deep regime (1237 J kg−1) and Cong regime (1111 J kg−1) significantly exceed the value for the ShCu regime (671 J kg−1)."
  **Answer:** Thank you for the comment. We modified the text as suggested.

- Line 258: CWP should be CWV
  **Answer:** done

- Lines 259-260: Combine these 2 sentences. "By the end of the day, the accumulation of water vapor in the column is negligible. This term is also nearly zero during Cong days."

**Answer:** Everything mentioned previously relates to the shallow regime description. Hence, it would not be reasonable to apply the same conclusion to both regimes unless the entire paragraph is readjusted.

- Lines 261-262: Change "which might affect the convective regimes developing the following day."

  To e.g.,:

  "which might act as a positive feedback for the continuation of nocturnal deep convection into the following day after Deep regime conditions"
  **Answer:** done

- Lines 265-266: Change "There is a characteristic low-level jet during the wet season. Anselmo et al. (2020) also reported an Amazonian low-level jet, occurring 10-40% of the time during March-May 2014-2015."

  To e.g.,:

  "The wind profiles for all regimes peak between the 900-800 hPa layer, characteristic of a low-level jet also reported by Anselmo et al. (2020), which they observed 10-40% of the time during GoAmazon between March-May 2014-2015."
  **Answer:** done

- Lines 265-272: Avoid describing relative wind speed maxima of a point sounding as "jets".
  **Answer:** Thank you for pointing out that inconsistency. Please note that we are using the 3-hour mean large-scale wind field from the VARANAL data instead of radiosonde data. For the low-level wind, we have cited Anselmo et al. (2020) as noted in the previous comment.

  Regarding the upper-wind, we only noticed one reference to the maxima as being the jet, on line 271. We modified this as:

  (...) upper-level  maxima. (...)

- Line 286: Word choice of "important subsidence"
  **Answer:** replaced by "robust subsidence"

- Lines 288-289: Remove sentence beginning "However, solely through differences in…"
  **Answer:** done

We hope these answers provided clarifications and answers for the comments raised in this revision. Below you can find the papers mentioned explicitly in our response.

References:

[1] Atmospheric Radiation Measurement (ARM) user facility. 2013. Meteorological Measurements associated with the Aerosol Observing System (AOSMET). 2013-12-12 to 2015-12-01, ARM Mobile Facility (MAO) Manacapuru, Amazonas, Brazil; MAOS (S1). Compiled by J. Kyrouac, S. Springston and M. Tuftedal. ARM Data Center. http://doi.org/10.5439/1984920.

[2] Atmospheric Radiation Measurement (ARM) user facility. 2014. Rain Gauge (RAINTB). 2014-10-14 to 2015-07-03, ARM Mobile Facility (MAO) Manacapuru, Amazonas, Brazil; Supplemental Site (S10). Compiled by J. Kyrouac, Y. Shi, M. Jane and D. Wang. ARM Data Center. http://doi.org/10.5439/1224827.

[3] Atmospheric Radiation Measurement (ARM) user facility. 2014. Laser Disdrometer (LD). 2014-09-24 to 2015-08-13, ARM Mobile Facility (MAO) Manacapuru, Amazonas, Brazil; Supplemental Site (S10). Compiled by D. Wang, M. Bartholomew, Z. Zhu and Y. Shi. ARM Data Center. http://doi.org/10.5439/1973058.

[4] Schiro, K. A., J. D. Neelin, D. K. Adams, and B. R. Lintner, 2016: Deep Convection and Column Water Vapor over Tropical Land versus Tropical Ocean: A Comparison between the Amazon and the Tropical Western Pacific. J. Atmos. Sci., 73, 4043–4063, https://doi.org/10.1175/JAS-D-16-0119.1.

---

## Author Comment (AC2)

**General Response to Referee #2**

Original referee comments are in blue.
Our responses are in black with regular font format. Text from the updated manuscript:

*Appears in italics with 1⁄2 inch indentation with the removed parts exhibited with a  and red color while new text added is shown in green.*

**Summary of RC2 comments:**
The paper is well-structured, and its results are effectively contextualized by comparing them with existing literature. Although certain analyses appear similar to prior studies, the utilization of different data and methodologies enhances the significance of this manuscript, particularly given the challenges of generalizing in-situ atmospheric observations. Notably, the exploration of moisture convergence analysis with this dataset appears to be novel.

We are grateful for the encouraging comments and would like to acknowledge the thoughtful revision, which helped improve our manuscript.

However, concerns persist regarding the potential impact of measurement uncertainties on the outcomes, as well as the statistical significance of the findings. (1) One of my main concerns regarding the methodology employed in this study, is the impact of the errors in the retrieval of the water budget terms, on the calculation of the divergence term.

This is addressed in responses 1-3 from the RC1. Here we are pasting the same answer.

In order to provide uncertainty estimates with respect to the water budget, we followed the reviewers' suggestions and considered different sources of precipitation in our updated analysis. In addition to the surface AOSMET precipitation [1], we are now also using the tipping bucket [2] and laser disdrometer (LD) [3]. Figure 9 in the current manuscript will be replaced by the figure below, which displays the updated results.

[Figure]

**Figure 9.** Surface water balance for (a,d) ShCu regime, (b,e) Cong regime, and (c,f) Deep regime. The upper panels represent the rates of change (mm/day) for water vapor convergence (CONV), evaporation (EVAP), precipitation (PREC), and the time derivatives of column water vapor ($\partial_t CWV$, where $\partial_t = \partial/\partial_t$) and liquid water path ($\partial_t LWP$). The lower panels display the accumulated water amounts for each term in the water budget along the day (mm). Note that CWV and LWP changes rely on microwave radiometer observations, evaporation on eddy correlation flux measurements, and precipitation  is estimated utilizing different sources, namely, the aerosol observing system surface data, tipping bucket, and laser disdrometer. The water vapor convergence term is estimated as a residual in the water budget equation (Section 2.2).

For the rate amounts, the time derivative of CWV or LWP corresponds to the microwave radiometer (MWR) data source. Evaporation rate is calculated from the Eddy Correlation (ECOR) Flux Measurement. Precipitation is the mean composite from the AOSMET, tipping bucket, and laser disdrometer sources. The precipitation uncertainty is estimated from the standard deviation of the different sources. Mean convergence is estimated from the water balance closure, i.e., equation (2) at line 137 from the manuscript. The standard deviation of mean convergence is estimated from the standard deviation of the mean $\partial_t CWV$, $\partial_t LWP$, EVAP, and PREC, i.e.,

$$\sigma_m = \sqrt{\sigma_m(\partial_t CWV)^2 + \sigma_m(\partial_t LWP)^2 + \sigma_m(EVAP)^2 + \sigma_m(PREC)^2} \quad .$$

Mean accumulation amounts and their standard deviations are obtained analogously. Note that the convergence uncertainty is mainly attributed to the uncertainty of CWV. More importantly, the main message of our analysis remains unchanged despite the uncertainties included, i,e., shallow days are dominated by divergence while deep days by convergence. Finally, to provide additional information to the readers, we will also include the precipitation and convergence term associated with each data source in the Supplementary Materials, as indicated by the Figure below:

[Figure]

**Figure A1.** Accumulated surface precipitation (mm) for the aerosol observing system surface data, tipping bucket, and laser disdrometer are shown in the upper panels (a-c). The corresponding convergence (mm) term for each instrument is displayed in the lower panels (d-f). The ShCu, Cong, and Deep regime results are shown in the first, second and third columns, respectively.

With this improved analysis, we believe that all the uncertainties related to the water budget are resolved consistently, given the available observational data. Likewise, the results are statistically significant.

To include the updated version of the surface water balance, the current manuscript must be modified as follows:

**Line 115**:

     *Instead of using the S-band radar precipitation, the water balance analysis*  *utilizes a combination of rain-gauge source measurements to provide a more robust estimation of the mean surface precipitation and its uncertainties. Specifically, we use the surface AOSMET precipitation [1], tipping bucket [2], and a laser disdrometer (LD) [3].*

**Line 141-143**:

     Specifically, CWV and LWP are based on the MWR. ECOR latent heat flux data is utilized for estimating the evaporation term. Precipitation is  obtained utilizing different sources, namely, aerosol observing system data, tipping bucket, and laser disdrometer (Section 2.1).

**Line 248-250:**

     The water balance results are shown in Fig. 9, with the top panels showing the hourly average rate values (mm/day) and the bottom panels showing the corresponding accumulated values (mm). The separate precipitation and convergence associated with each rain gauge used to estimate the mean surface precipitation and its uncertainty in the water budget results are provided in fig. A1. First, we notice that the variation in LWP appears negligible, and it does not contribute significantly to the water budget.

**Line 254-255:**

     At day's end, the ShCu regime is estimated to lose 3.4 mm of water vapor due to divergence, while the Cong regime loses  0.9 mm. In contrast, the Deep regime gains  5.2 mm of water vapor due to convergence.

**Figure 9, page 14:**

     **Figure 9 is replaced by the updated one, shown above. To consider the new sources of surface precipitation, the caption is updated as follows:**

     Note that CWV and LWP changes rely on microwave radiometer observations, evaporation on eddy correlation flux measurements, and precipitation  is estimated utilizing different sources, namely, the aerosol observing system surface data, tipping bucket, and laser disdrometer.

(2) The need to define a control volume in the water budget.

Our surface water balance analysis is based on the integral form of the total water continuity equation. While a control volume is only properly defined in models or theory, we can still define a scale where our analysis can be applied, as follows:

We only employed local surface measurements in the water budget with the intention of conducting a local surface water budget analysis. The term local is loosely used to indicate that the water budget is suitable for a local analysis, which could be imagined as a scale of ~ 1-10 km or single cloud size scale. However, the scales depend on each instrument. Surface fluxes depend primarily on the surface properties, and should not vary much at different points over the same vegetation type, but there is only one ECOR observational site for surface measurements. The infrared thermometer in the MWR has a field of view of less than 3 degrees, hence the column water vapor measurement represents the value over the site. Gauge-measured precipitation is a point measurement and can vary a lot over short distances in individual rain events. However, since an hourly average is applied to these data and our composites average over many events, these point measurements can characterize the scale of the event, i.e., of the clouds that are advected over the vertical experimental site.

To include this discussion in the manuscript, the following modifications were made:

**Line 118-119:**

Hourly averages are applied for the data utilized in the water budget analysis, and its mathematical derivation is provided in the next section.

**Line 143-144**:
The water vapor convergence term is estimated using the mean composites for $\partial CWV/\partial t$, $\partial LWP/\partial t$, EVAP, and PREC.
Note that we only employed local surface measurements. Surface fluxes depend primarily on the surface properties, which are uniform around the experimental site, at least up to 10 km away. Surface rain gauges and the microwave radiometer perform point measurements. However, since we are applying hourly averages for the water budget datasets and the composites average over many days, the mean precipitation, CWV, and LWP relate to regions associated with the domain average of a cloud size scale. Therefore, our water balance is consistent for a temporal-spatial scale of ~1 hr and ~10 km.

(3) Discuss the impact of the hydrostatic approximation in the calculation of CWV and LWP in the water budget.

We have assumed the hydrostatic balance only to perform the mixing ratio integration for the radiosonde measurements (Figure 7). Although equation (1) indicates integration at pressure levels, the column water path and liquid water path were taken from the microwave radiometer ([https://www.arm.gov/capabilities/instruments/mwr](https://www.arm.gov/capabilities/instruments/mwr)) for the water budget analysis. As reported on the ARM MWR website, integrated water vapor and liquid water paths are derived from **radiance measurements** with a statistical retrieval algorithm that uses monthly derived and location-dependent linear regression coefficients. The MWR calibration does not involve the hydrostatic approximation.

Nevertheless, the MWR does not provide optimal measurements during precipitation events. This issue has a particular impact on the Deep regime composite, which also tends to deviate further from hydrostatic balance. Overall, the hydrostatic balance approximation would not affect the entire column mixing ratio (but near convective clouds) and the hourly averages in the water budget may also help minimize the impacts of any errors associated with unbalanced conditions. Moreover, the primary sources of error in the CWP and LWP within the water budget are due to the accuracy of the MWR and its limitations in operating under precipitation conditions, although data were removed during rainy periods, as we explained:

Lines 116-118: Here, CWV and liquid water path (LWP) are taken from the microwave radiometer (MWR). Similarly to Schiro et al. (2016), we exclude data in cases where the brightness temperature surpasses 100 K, and when water accumulates on the MWR lens surface during rainy periods.

Note that the primary source of uncertainty in the water budget is attributed to the CWV uncertainty across all convective regimes. Therefore, the error accounted for in the water budget serves to estimate the deviation from hydrostatic balance.

(4) Discuss the influence of the small sample sizes for regime analysis.

Our study encompasses 60 deep days, 27 congestus days, and 16 shallow days during the wet season, which is a sample size similar to other studies in the literature that also used GoAmazon data (Ghate & Kollias 2016, Zhuang et al. 2017, and Tian et al. 2021) and a much larger sample than studies that preceded the GoAmazon experiment (Machado 2002, Betts 2002, Silva Dias 2008, Khairoutdinov and Randall 2006, etc…).

Nonetheless, we agree with the reviewer that given the variability of clouds, convection, and precipitation, the larger the sample the better. Nevertheless, we can still make relevant scientific inferences from this data as long as we consider all uncertainties. In

that regard, following the recommendations of both reviewers, the updated manuscript now includes estimates of the error in the water budget analysis (answers 1-3 for RC1 and answer 1 for RC2), and error bars have been added to all the remaining figures.

Having said that, we note that having a larger dataset for clouds and convection requires a huge scientific effort to make these measurements over tropical forests. This is now being pursued by Brazilian scientists at the ATTO tower. Before long, our scientific community will have access to a more extensive and comprehensive dataset for studying clouds and convection than what the GoAmazon campaign has provided.

(5) To Include an instrumentation table in the methodology section with the name and short description of each data source.

Ok. We have included the following paragraph at the end of the data section:

Line 126:
Finally, a table summarizing the data is also provided in the supplementary material (table A1).

We also noted that the microwave radiometer citation was missing. The text was updated as follows:

Line 116:
Here, CWV and liquid water path (LWP) are taken from the microwave radiometer (MWR, ARM, 2014d)

The table included in the manuscript is shown below:

**Table A1.** Data

| Name | Description |
|------|-------------|
| Cloud mask (Giangrande et al., 2017) | Combines multiple sources of cloud data to provide a high-resolution temporal (30 sec) vertical profile of cloud type, including shallow, congestus, and deep. Cloud frequency profile is also calculated as the cloud counting fraction over 12 minutes. |
| S-band radar (Schumacher and Funk, 2018a, b) | Volume data of reflectivity and derived precipitation rate at 2 km. Rain cover is calculated as the fraction of reflectivity $> 20$ dBZ over a 100x100 km$^2$ analysis domain. These data have a temporal resolution of 12 minutes, a horizontal resolution of 2 km, and a vertical resolution of 500 m. |
| Surface precipitation | We employ three source of surface precipitation data, namely, aerosol observing system surface data, tipping bucket, and laser disdrometer. They have only a temporal dimension, for which we use an average over 12 minutes or 1 hour. |
| Atmospheric State | Radiosonde data has a latency of 6 hours (launches at 02, 08, 14, and 20 LST). We utilized sounding profiles covering at least 8 km of the atmosphere. A derived planetary boundary layer height is also used. Additional variables, such as CWV, MLCAPE, and MLCIN are calculated. |
| Surface fluxes | Hourly average data from the ARM best-estimate dataset based on observations from the Eddy Correlation Flux Measurement. Surface evaporation is calculated and employed in the water budget analysis. |
| Water content | Hourly average data of CWV and LWP taken from the microwave radiometer are employed in the water budget analysis. |
| Large-scale wind field | Based on the variational analysis. It corresponds to a 3-hour average and a domain average of $\sim$100 km. |

**(6) The arbitrariness of the 2% threshold for 'rain coverage,'**

This particular threshold is based on [4], as we indicated in the methods section. The regime classification method is inherently somewhat arbitrary. All the definitions (i-v, lines 157-167) are consistent, but not unique. We conducted numerous tests to develop our methods, including different cloud boundaries, minimum time frequency for observed clouds, among others. It is not feasible to show all the tests. However, we noticed that the mean composite of each analyzed variable always showed similar behavior. Therefore, small changes in the regime methods will cause small changes in the results, but qualitatively the interpretation of each result is unchanged due to these arbitrary possibilities. Given that, our final decision to choose the method definitions was based on closely following previous studies and prioritizing simplicity.

For instance:
- Cloud boundaries are based on Giangrande et al. (2017).
- Rain cover definition is based on Zhuang et al. (2017), which manually tested several parameters. Their results indicated that shallow rain cover never exceeds

2%, a criterion we adopted in our definition of minimum rain cover for identifying congestus or deep clouds.

- Early morning perturbation is based on Tian et al. (2021). However, they requested a precipitation rate of less than 0.015 mm/hr between 08:00 and 09:30 LST, whereas we adapted the procedure by requiring no observation of congestus or deep clouds between 06:00 and 10:00 LST.
- Local convection definition are based on the spatial scale of mesoscale convective systems, which typically cover approximately ~100 km [5]

Since every convective classification procedure is based on previous studies, this further justifies all the chosen parameters.

(7) The definition of 'bulk wind shear' is somewhat arbitrary.

We used the standard definition of bulk shear, as seen, for instance in reference [6], i.e.,

$$\text{wind-shear}(z) = \sqrt{[u(z) - u(0)]^2 + [v(z) - v(0)]^2}$$

The reviewer suggested that calculating the bulk shear from the level of maximum wind to the surface would complement our analysis with additional information, such as the height of the given maximum. We conduct the corresponding analysis by taking the maximum wind speed below 4 km (≈600 hPa), a suitable condition to evaluate only the low-level jet. The results are shown in the figure below.

[Figure]

**Figure A2.** (a) Vertical bulk wind shear from the level of maximum wind speed below 4 km and (b) the associated pressure level.

Comparing the above results to our standard analysis (Figure 13, pag 17), it indicates that the main message still remains, where deep days exhibit the strongest wind shear only in lower levels. For the associated height levels for maximum wind speed, the same analysis that we conducted using figure 10 (wind speed profile, pag 15) also holds (lines 267-269).

Lines: 267-269
*"During the morning, the ShCu regime reveals a lower and slightly stronger jet. However, the PBL grows during the day to a height of 1-2 km (see Fig. 8), reaching higher altitudes for ShCu days. As a result, the lower jet in the ShCu regime is more significantly affected by the PBL growth."*

However, the following manuscript sentence must be improved:

**Lines 269-270:**

*Thus, at 14 LST, the ShCu regime reveals a less prominent low-level wind peak but at a slightly higher altitude compared to Deep days.*

Therefore, both properties of height related to maximum wind speed and wind shear can be similarly obtained in figure 10 and figure 13, respectively. However, we included fig. A2 to further support our analysis. Thus, the text of the manuscript has been updated as follows:
Lines 290-293:
To evaluate the vertical wind shear, we used the bulk wind shear, which is defined as the magnitude of the vector difference of the wind at two levels. Figure 13 shows the vertical bulk wind shear for the layers 0-2 km, 0-4 km, and 0-6 km. Additionally, the bulk shear from the level of maximum wind speed below 4 km and the associated pressure level is provided in fig. A2. The 0-2 km layer exhibits a greater dependence on the diurnal cycle, with the Deep days followed by Cong days showing the most substantial wind shear at any time. Moreover, the Deep regime shows the largest difference between the maximum wind speed below 4 km and surface wind.  These results suggest that low-level vertical wind shear is related to convection development.

Figure A2 above is also included in the supplemental materials.

(8) Additionally, there also have been a few technical corrections reported regarding the text, aimed at enhancing sentence structures.

- line 14: Missing comma after 'even moisture'
  **Answer**: done

- Line 46: Define 'mid-levels', for clarity
  **Answer**: We usually apply this term in a loose way. Here, the text is updated by "a moister layer from the surface to mid-levels (~ 5 km)"

- line 72: Typing error: 'a city border the Rio Negro'
  **Answer**: I didn't understand. Rio Negro is the portuguese name of "Negro River." I replaced a city border with a city that borders if that you mean by typing error.

- line 77-78: Suggested: '... a cloud mask based on time-height profiles of the cloud location'
  **Answer**: done

- Line 83: What do you mean by 'cloud frequency profile'? Relatedly, in Fig 5's caption, what is the percentage cloud frequency relative to?
  **Answer**: The cloud-mask gives one cloud type at every 100 m in the vertical and every 30 seconds, which are the space and time resolutions of the cloud mask data product developed by Z. Feng and S. Giangrande (Giangrande et al., 2017). The cloud types that we are considering are: shallow, congestus, and deep. The deep cloud frequency, for example, is the number of deep clouds found in a 12 min interval divided by 24 (the maximum possible) at each height. .

  To make this more clear, the text in the figure caption is updated as follows: "Cloud frequency (cloud counting fraction in %, colormap) as a function of height calculated over a 12 minutes window based on the cloud mask and surface precipitation rate (mm/hr, red line) locally measured by the aerosol observing system at the T3 site."

- line 107: 'water vapor mixing ratio'
  **Answer**: done

- Line 112: Do you mean 'the 100-hPa depth layer immediately above the surface'?
  **Answer**: Yes. We checked, and the methods explains it on Line 112:
  Line 112: A mixed-layer parcel **from the surface to 100 hPa** is used as the initial state for the parcel's ascent Stull (2016).

- Line 126: Need to define the control volume over which the conservation equation is being applied.
  **Answer**: This is addressed in the answer (2).

- Line 129: Please rephrase for clarity: 'E and P correspond to the water mass fluxes associated with **surface evaporation** and precipitation'
  **Answer**: Text modified as suggested.

- line 131: 'with the divergence of water vapor'
  **Answer**: done

- Line 132: I believe this statement is incorrect. Since the variable here is total water mass mixing ratio, it is not affected by phase transitions associated with cloud formation in the control volume.
  **Answer**: Thanks for catching that misleading statement. Looking at fig. 9, we clearly see that convergence and precipitation dominate the water budget during the cloud deepening period (panel c, 16-17 LST). Therefore, this phrase was removed:

  **Line 131-133:** The second integral is the water mass flux divergence, which is mostly associated with water vapor.

- Line 134: I believe there is a mistake here. The terms in Equation 1 have units of mass flux, such as kg/m²/s. Dividing by the density of liquid water would give units of m/s, which does not seem to be the intention. Moreover, to express the equation in terms of liquid and vapor water paths, it is not necessary to divide by anything, since the integral of the total water mixing ratio divided by the gravity acceleration with respect to pressure already represents the total water path in a hydrostatic atmosphere. This total water path can then be separated into vapor, liquid, and ice terms, as you mention below.

  **Answer**: This comment was just explaining about the change of units. We have to divide by the density of liquid water to get mm-of-rain (height of the equivalent layer of water on the surface) instead of kg/m2 (which could be from ice, liquid, or vapor).

  To make this more clear, we modified the text as follows:

For the sake of this analysis, we  neglect the time variation of the ice term and express all terms in units of mm hr$^{-1}$.

- Line 134: 'ignore' should be 'neglect'
  **Answer**: done

- Line 134: Are you neglecting the time variation of the ice mass or the presence of ice itself in the analysis? Does the observed LWP you use include all condensate or does it discriminate between ice and liquid water?
  **Answer**: Based on the MWR paper (https://ieeexplore.ieee.org/document/4373386), they use ground-based two-channel microwave radiometers to provide observations of downwelling emitted radiance from which precipitable water vapor (PWV) and liquid water path (LWP) are retrieved. Thus, our measurements of LWP do not include ice. The comment above is correct, i.e., the ice water path is neglected.

  The IWP is only retrieved by the upgraded version of the MWR. The MWR-Profiler (e.g., MP-3000 from Radiometrics) has more channels, has a wider field of view (30deg instead of 3o), and can retrieve the vertical profiles of temperature and humidity, as well as LWP and IWP. However, there are many greater uncertainties in these retrievals and those from the standard MWR.

- line 138: If you follow the suggestion I mentioned earlier and avoid dividing by any additional factors, the terms in the equation can be represented as EVAP = E and PREC = P. I recommend maintaining one naming convention for consistency and clarity
  **Answer**: We are using EVAP and PREC as rates (mm/hr).

- Line 141: The data in Fig 9 seems to have a higher frequency, intervals seem to be of 12 min.
  **Answer**: The water budget analysis is based on hourly averages. This is discussed in answer (2). Hence, the scatter plot shows 1-hr intervals.

- Line 152: Do these times define what you call 'diurnal cycle' throughout the paper? If so, please clearly define the term 'diurnal cycle' when it is first used in the text, to ensure that readers understand the time period being discussed.
  **Answer**: The diurnal cycle is first defined for the first time in his line. It refers to the period 10-20 LST. It is also mentioned in lines 157-159.

- Line 153: By 'rain coverage' are you referring to large hydrometeors, independent of the phase. Please clarify.

**Answer**: The rain coverage is calculated as the fraction of reflectivity pixels > 20 dBZ over the analysis domain (100x100 km$^2$ , centered at T3). This is a common threshold, e.g., utilized in [4].

Based on reference [4]: "The 20 dBZ and 40 dBZ rain fractions, defined as fractional coverage of **Z ≥ 20 dBZ representing the area of convective systems** and Z ≥ 40 dBZ representing the area of intense precipitation or a convective core in the 100 km grid box, are calculated for each level."

- Line 157-164: By 'precipitation', do you mean 'rain coverage'? If so, please correct the text here for consistency.

**Answer**: Exactly, thanks for pointing that out. We have modified the text:

In criteria (i-iii),  is replaced with Rain cover.

- Line 165-166: Above, you list the conditions for each type of convection regime, while here you list conditions that apply to all regimes, right? If so, I suggest clarifying this.

**Answer**: Line 165: iv) Early morning perturbation: For the 06-10 LST period, we require that no congestus and deep clouds be observed.
The criteria iv motivation is given in lines 169-171: "The exclusion of any relevant early morning disturbance (06-10 LST) associated with important pre-convective activity guarantees that nighttime MCSs do not cause significant preconditioning."
The criteria v is only about local convection, i.e., a filter to remove MCSs in our convective regime classification.

- line 173: I suggest 'the propagating-convection category occurs more frequently during the wet season', for clarity.
**Answer**: done

- line 195: I suggest rewording 'because' to: 'associated with'.
**Answer**: done
" associated with both the boundary layer height and the lifting condensation levels  being lower in the Deep regime."

We updated Fig. 6 to limit the y-axis from the surface to 5 km and also included the estimated standard deviation for the difference in potential temperature and mixing ratio

$$\left(\sqrt{\sigma_m(Deep)^2 + \sigma_m(ShCu\ or\ Cong)^2}\ \right),\ \text{as shown in figure below.}$$

[Figure]

**Figure 6.** Atmospheric conditions. (a) Potential temperature (K) and (b) water vapor mixing ratio (g/kg) at 08 LST radiosonde observations. The corresponding convective regime differences (Deep-ShCu and Deep-Cong) for potential temperature ($\Delta\theta$, c) and mixing ratio ($\Delta r_v$, d) are also included.

Despite the uncertainties in the difference of mixing ratio between Deep and ShCu composites, values are greater than 0 from the surface up to 3 km. Hence, our main message remains the same: Deep days are relatively moister than shallow days only in the lower levels. The differences between congestus and deep days are less

pronounced, albeit the deep composite tends to be slightly moister than the congestus profile.

- line 214: I suggest providing a quantitative measure instead of 'is quite similar'. Maybe something like: 'the difference between the moisture profiles for different regimes is less than 1/x of the maximum difference at lower levels'.
  **Answer**: Based on the updated figure (containing errors), discussing the values above 3 km is not advisable due to statistical errors. The text was updated as follows:

  Previous version:

  Updated version:
  Above 3 km, the differences in moisture profile for all regimes have little statistical significance.

- line 217: Suddenly starting to talk about the tropical oceans reads a bit strange. If the goal is to mention contrasting results, I suggest rephrasing it to smooth the transition, for example: 'Moreover, these results contrast with studies over tropical oceans, where free-tropospheric humidity has been shown to play a more significant role...'.
  **Answer**: Thanks for the observation. We modified the text as suggested.

- Line 223: Where it says '...at 2 LST', it should be noted that since the classification was done for times between 10-20 LST, measurements at other times of the day might include different types of cloud than those in the regime in question.

  **Answer**: Based on the RC1 technical comment, we modified the text above as follows:

  Lines 222-224:  The differences in CWV among Cong and Deep regimes show little statistical significance from nighttime to early morning, while their difference is maximum at 14 LST when it reaches 2.1 mm.

Since we mention that the difference in CWV between Cong and Deep regimes before the diurnal cycle has little statistical significance, it no longer seems appropriate to include observations about the methods during nighttime.

- Line 231: Where is says 'despite slightly larger latent heat', add a reference to the section of the manuscript where this is shown.

   **Answer**: Thanks for the suggestion. The text is updated as follows:
   "despite slightly larger  surface evaporation (section 4.3)"

- Line 233: 'water vapor convergence', ditto

   **Answer**: The text is updated as follows:

   "water vapor convergence (section 4.3)"

- line 242: Please maintain the naming convention (MLCAPE for CAPE).
   **Answer**: done. However, sometimes we use CAPE to refer to other studies. In this case, the CAPE notation was maintained.

- Line 249: It appears that you are calculating 'accumulated values' by numerically integrating the mm/day values over time intervals of approximately 12 minutes. However, it's unclear how the resulting values are still being expressed in mm. Given that the time step (dt) is 1/120 days, it seems that the resulting unit should be mm multiplied by the time step, which would be mm/120.

   **Answer**: The top panel shows the variables as a function of time, in units of mm/day. The bottom panel shows the same variable but integrated in time since mid-night (local time), hence the time units cancel. The 120 factor is already included in the computation, when we convert minutes to days, as the integration is a sum of terms multiplied by dt:

   $$accum\ precipitation\ =\ \int_{0h}^{24h} precip(t)dt$$

   Precip is mm/day, accumulated precip is mm, and dt = 0.00833 days == 0.2 hours == 12 min.

- line 249: Instead of 'variation in LWC', I suggest using '$\partial t$LWC', for clarify
   **Answer**: done

- line 250: 'dominated' seems to imply that CONV is the largest term among those in the budget analysis, so I would suggest rewording, for example, 'shows mostly water divergence'.
  **Answer**: done. Note that this sentence is also modified due to the next comment.

- Line 251: The 'divergence' term responds to changes in the total water, not just water vapor. I understand that the convergence is expected to occur in the form of water vapor, since horizontal transport of condensate is less likely, but this should be somehow clarified, at least in a footnote.

  **Answer**: Yes. To emphasize that divergence responds to changes in total water, we modified the text as follows:

  Previous text:
  Lines 250-251:  However, the water budget closure requires convergence in the period from 14 to 18 LST.

  Updated text:
  The daytime of ShCu and Cong days shows mostly water vapor divergence. This is primarily due to more significant negative changes in CWV and lower precipitation rates, whereas changes in condensate exert a minor influence on the water budget. On the other hand, the Deep regime exhibits relatively neutral conditions from night to early afternoon. However, the water budget closure requires convergence in the period from 14 to 18 LST.

- line 251: Suggest: 'relatively neutral'
  **Answer**: done (see text above)

- Line 256: I suggest framing the discussion slightly different: instead of saying that high evaporation balancing the divergence leads to low precipitation, it is seems more intuitive to think it as high evaporation and low precipitation requiring strong divergence for closure.

  **Answer**: Thanks for the suggestion. The text is updated as follows:

  Previous version:

Updated version:
For ShCu days, relatively high surface evaporation and the absence of precipitation require a strong divergence for water balance closure. The Cong regime shows a significant divergence but is relatively weaker compared to the ShCu regime, as their surface evaporation is relatively similar, and congestus precipitation exhibits modest values.

- Line 257: Ditto

    **Answer**: Addressed above.

- line 258: Is CWP a typo? Do you mean CWV? As commented before, it is clearer to say $\partial_t$CW V than 'the variation in CWV'.
    **Answer**: done
- Line 259: What do you mean by 'accumulation of water vapor'? Are you referring to Fig. 9d? If so, please clarify. Similarly, clarify the panels you are referring to in the rest of the text.

    **Answer**: Accumulation always refers to the lower panels (given in mm). Figure 9 caption indicates this: "The lower panels display the accumulated water amounts for each term in the water budget along the day (mm)." Nevertheless, we updated the text as follows:

    Lines 258-259: For ShCu days, the variation in CWV tends to be small and negative from nighttime to early morning. Then, it increases and balances around noon. By the end of the day, the accumulation of water vapor (fig. 9d) in the column is negligible.

- Line 269: Please expand briefly on how the LLJ is affected by the PBL growth. Do you mean that the PBL growth slows down the LLJ? Please clarify that this is one possible explanation, since other factors may be involved in regulating the intensity of the LLJ.
-
    **Answer**: The factors contributing to peaks in low-level wind speed can vary significantly depending on the specific geographical context. However, several

primary dynamic mechanisms associated with low-level jets include inertial oscillation driven by diurnal changes in eddy viscosity, as well as terrain effects such as the formation of slope and valley winds or induced wind deflection. Nevertheless, there is currently a lack of literature addressing the occurrence of a large-scale low-level jet within the Amazon. Given this, our intention is not to fully explain the mechanism associated with the low-level jet, but rather to provide a simple rationale justifying the observed differences among the different regime categories and the PBL, which is another result described in the manuscript. In light of this, we have included an additional sentence to address the reviewer's suggestion.

Lines 267-269: During the morning, the ShCu regime reveals a lower and slightly stronger jet. However, the PBL grows during the day to a height of 1-2 km (see Fig. 8), reaching higher altitudes for ShCu days, and the mixing of free-tropospheric and PBL momentum potentially reduces the wind speed. As a result, the lower jet in the ShCu regime is more significantly affected by the PBL growth.

- Line 271: Better refer to the layer between ~ 600 and ~ 350 hPa or so, since at 300 hPa the wind tends to be intense in the deep case.

  Answer: The text was updated as suggested:

  "For the mid- and upper-levels, between 600-300 600-350 hPa layer, the ShCu regime shows an additional upper-level jet, while the Cong and Deep regimes exhibit weaker and comparable wind speeds."

- line 275: Suggested: 'The most notable difference'
  Answer: done

- Fig 9, caption: Instead of referring to the number of the subsection, I suggest referring to the number of the equation.
  Answer: done

- Line 295-297: Although the 6-km level is taken as a reference here, the corresponding 'bump' in the wind profile starts at 4km and reaches 8km height (as you mentioned in the discussion of Fig. 10). Moreover, it would be incorrect to say that only because it happens at levels higher than the top of the cloud, the wind pattern does not matter for the convection development. For example, the wind profile impacts the gravity wave pattern induced by convection itself, and

these waves may then feedback on the dynamics of the convective clouds [e.g. CHK86].

**Answer**: Considering the reviewer's comment, the following text in lines 295-297 is removed:

- line 308: 2.5 J/g = 2500 J/Kg, not 250 J/kg. But there is no need to mention both values anyway

**Answer**: OK. We did not make any modifications. In Fig. 14d, we converted the unit for visibility because the x-axis is compact, making smaller numbers more suitable for use.

- Line 321: Based on this results, you can only state that CAPE is not a good indicator for precipitation. This analysis doesn't involve convection strength, but precipitation only.

**Answer**: Agreed. The manuscript text is updated as follows:

"These results suggest that CAPE is not a good indicator of  precipitation in the Amazon, which is consistent with the findings of Itterly et al. (2016); Schiro et al. (2018)."

- Fig 14, caption: 'average' should be 'averaged'. Also, what do the vertical bars mean?

**Answer**: OK, words changed as suggested.
Vertical bars correspond to the standard deviation of the mean variable.

- Line 328: 'Results showed that isolated deep convection is associated with more extensive and longer-lived clouds throughout the diurnal cycle.' Do you mean compared to shallower clouds? I suggest omitting this statement, as such feature can be considered 'common knowledge'.

**Answer**: done

- Line 339: Please clarify what you mean by 'noted significant differences between morning atmospheric conditions and convective intensity'.

**Answer**:

The phrase was too long and difficult to understand. What we wanted to say here is that Itterly et al (2016) observed significant correlations between the morning conditions of different variables and the convective intensity (it is stated in their abstract).

We change the phrase as follows:

Our results show that deep days are associated with moister conditions in the early morning, but only in the lower troposphere, particularly below 3 km. This contrasts with the results from Itterly et al. (2016).  They also noted significant differences between morning atmospheric conditions and convective intensity, but indicated that   humidity in the upper  troposphere also exhibits  a strong relationship with convective intensity.

- line 366: It should be 'captured', instead of 'capture'.
   **Answer**: done

- Line 380-382: Did they suggest that wind shear would both favor and hinder convection? Please clarify.

   **Answer**: The authors relate wind shear intensity with convection intensity. Deep days show the strongest wind shear at both layers 0-3 or 0-6 km only in the dry season, thereby favoring convection. However, the authors observed that Deep days do not show the strongest wind shear in the 0-6 layer during the wet or transition season, thereby suggesting that vertical wind shear could have no impact or might hinder convection in those seasons. There is no explanation in their paper relative to this specific interpretation. Please, check their paper [4] for more information. It is the last sentence before Conclusions and Discussion:

   > *"Thus, while larger vertical wind shear may appear to link to the shallow-to-deep convection transition during the dry season, it may have no influence or suppress the shallow-to-deep convection transition during the wet and transition seasons."*

   Line 406: 'this value', which one?

**Answer**: Vertical velocity. The text is updated as follows:

"Nevertheless, it should be noted that  vertical velocity is challenging to assess through observations."

We hope these answers provided clarifications and answers for the comments raised in this revision. Below you can find the papers mentioned explicitly in our response.

References:

[1] Atmospheric Radiation Measurement (ARM) user facility. 2013. Meteorological Measurements associated with the Aerosol Observing System (AOSMET). 2013-12-12 to 2015-12-01, ARM Mobile Facility (MAO) Manacapuru, Amazonas, Brazil; MAOS (S1). Compiled by J. Kyrouac, S. Springston and M. Tuftedal. ARM Data Center. http://doi.org/10.5439/1984920.

[2] Atmospheric Radiation Measurement (ARM) user facility. 2014. Rain Gauge (RAINTB). 2014-10-14 to 2015-07-03, ARM Mobile Facility (MAO) Manacapuru, Amazonas, Brazil; Supplemental Site (S10). Compiled by J. Kyrouac, Y. Shi, M. Jane and D. Wang. ARM Data Center. http://doi.org/10.5439/1224827.

[3] Atmospheric Radiation Measurement (ARM) user facility. 2014. Laser Disdrometer (LD). 2014-09-24 to 2015-08-13, ARM Mobile Facility (MAO) Manacapuru, Amazonas, Brazil; Supplemental Site (S10). Compiled by D. Wang, M. Bartholomew, Z. Zhu and Y. Shi. ARM Data Center. http://doi.org/10.5439/1973058.

[4] Zhuang, Y., R. Fu, J. A. Marengo, and H. Wang (2017), Seasonal variation of shallow-to-deep convection transition and its link to the environmental conditions over the Central Amazon, J. Geophys. Res. Atmos., 122, 2649–2666, doi:10.1002/2016JD025993.

[5] Houze, R. A.Jr. (2004), Mesoscale convective systems, Rev. Geophys., 42, RG4003, doi:10.1029/2004RG000150.

[6] Stull, R., 2017: "Practical Meteorology: An Algebra-based Survey of Atmospheric Science" -version 1.02b.  Univ. of British Columbia.  940 pages.  isbn 978-0-88865-283-6 .

---

## Referee Report (RR1)

**Review of Viscardi et al. 2023, ACP**
**"Environmental Controls on Isolated Convection during the Amazonian Wet Season"**

The authors made an effort to address the comments and suggestions from the initial review, and the revised manuscript shows improvement over the original version. However, further revisions are required, as outlined below.

Note: Suggestions to explain or expand on certain aspects refer to the manuscript itself, not just the authors' response.

**1   Major comment:**

Making inferences about a population requires a rigorous assessment of the statistical significance of the results obtained from a sample. In other words, smaller samples increase the likelihood that random errors in observations could explain the observed behavior. Applied to this study, this means that ensuring the differences between shallow, congestus, and deep convection regimes seen in the sample can be generalized requires an understanding of how likely these differences are to occur simply by chance.

After being asked to include a discussion on the implications of the relatively small sample size for their results, the authors argue that they have considered all uncertainties, while relying on the sample's standard deviation as an indication of "error." However, the standard deviation of the sample only informs about the distribution of values within the sample and does not necessarily indicate how close the sample mean is to the population mean. To assess this, an evaluation of the standard error, which is related to the standard deviation of the sampling distribution of the mean, is needed.

Increasing the sample size may not be feasible nor within the scope of the paper, but discussing the limitations of the results obtained from this sample would significantly improve the conclusions of this study. Therefore, my recommendation is for the authors to apply a bootstrapping or other resampling technique to this dataset to better support their conclusions and account for the potential impact of the sample size on the results.

**2   Minor comments:**

1. To ensure completeness, it is suggested that the authors include the error propagation formula for the CONV term in the budget equation, as mentioned in their response, within the manuscript itself. This formula involves implicit assumptions, such as a lack of correlation among the terms $\partial_t \text{CWV}$, $\partial_t \text{LWP}$, EVAP, and PREC, which may not be immediately apparent. Explicitly stating these assumptions should enhance the clarity of the manuscript. Similarly, the authors should provide an explanation of how the standard error of the PREC term was obtained from the three considered data sources. Including this information will further improve the understanding of the methodology employed in the study.

2. The manuscript contains multiple statements based on speculation, which are acceptable since, in most cases, observations cannot provide measures of all elements potentially affecting the behavior of certain variables. However, such statements must be treated as speculative by including qualifiers such as "likely" or "presumably" instead of making assertions without proper evidence. While some instances of such statements were indicated in the previous review, similar

expressions can still be found in the updated manuscript. It is the authors' responsibility to thoroughly address and revise these instances throughout the text. As a reference, the following statements (lines 148-152) serve as examples: "... are uniform around the experimental site", "CWV and LWP relate to regions associated with the domain average of a cloud size scale", "our water balance is consistent for a temporal-spatial scale of $\sim 1$ hr and $\sim 10$ km." In these cases, the authors should consider rephrasing the statements to accurately reflect the speculative nature of their claims.

3. The text on lines 223 and 233 uses the phrase "little statistical significance." It is advisable to avoid referring to the statistical significance of results unless they have been objectively analyzed through parametric statistical tests or resampling strategies. In such cases, the methods and outcomes should be explicitly clarified in the manuscript, as previously mentioned.

4. The authors have argued that the results remain consistent when using different rain coverage thresholds for the convection regime classification, other than the 2% threshold mentioned in the paper. While it may not be practical to present all possible analyses, mentioning this consistency in the text would add value to the discussions and support the robustness of the findings.

5. The comment made during the previous review round about the statement "A mixed-layer parcel from the surface to 100hPa is used..." might have been unclear. To clarify, it is suggested that this statement could be rephrased as "the 100-hPa layer immediately above the surface" to provide better context and understanding.

6. The authors have clarified that they neglect ice in their analysis; however, the manuscript still states "neglecting the time-variation of ice," which should be revised to "neglecting ice" for clarity. Additionally, the authors justify this neglect by stating that the time variation in LWP is negligible in the budget analysis. Since ice water path was not considered in the study, further clarification is needed to support this justification. It appears that the authors assume $\partial_t \text{IWC} < \partial_t \text{LWP}$ in general, but this assumption is neither evident nor explicitly mentioned in the manuscript. To ensure transparency, the authors should provide a justification for this assumption within the manuscript.

7. In Section 2.3, it is recommended to clarify the term "rain" in "rain coverage" to specify that it refers to radar echoes exceeding a given threshold, regardless of whether the phase is ice or liquid. This will provide better context for the readers and improve the understanding of the methodology.

8. In the caption for Figure 6, it would be helpful to provide clarification on how the error bars for panels c and d were obtained.

9. Line 320 and Fig. 14's caption: 2.5 J/g $\neq$ 250 J/kg.

10. The author's effort in explaining the interpretation of Zhuang et al. (2017)'s results in their response is appreciated. However, the original intent of the comment was to recommend revising the explanation within the manuscript (lines 390-393) for improved clarity and understanding.

11. Line 27: "deep convection in either..." should be "deep convection either..."

12. Line 62: It is recommended to use an alternative term for "sensitivity," such as "assess correlations," to more accurately describe the intended analysis and avoid potential misinterpretations.

13. Figure 1's caption: It is recommended to clarify the intended meaning of the phrase: "The dotted circle describes the S-band radar domain with *available measurements*.".

14. Line 107: It would be helpful to clarify the assumptions made in the vertical integration up to 200 hPa, in cases where the sounding only reaches up to 8 km.

15. Line 244: The term "mixed-layer MLCIN AND MLCAPE" contains redundant elements, as the "ML" prefix already indicates that these variables refer to the mixed-layer. Consider revising this to "MLCIN AND MLCAPE" for conciseness and clarity.

16. Line 261: The phrase "Changes in CWV" should be represented as $\partial_t$CWV for clarity. Regarding "changes in condensate," please specify if this refers to $\partial_t$LWP. Furthermore, it appears that "as" might be a more appropriate choice than "whereas," and "low" instead of "lower." Please review and revise the wording accordingly. It is important to note that negative CONV (or divergence) is not solely due to the mentioned factors, as EVAP is actually the dominant term. Please adjust the explanation to accurately reflect this information.

17. Figure 9: It is recommended to use a symmetric logarithmic scale for the y axis.

18. Line 282: The phrase "more significantly affected by PBL growth" can be reworded as "likely more influenced by PBL growth compared to other regimes" to enhance clarity and accurately convey the intended comparison.

19. Line 285: It is recommended to add "respectively", at the end of the phrase.

20. Line 309: The term "more robust" in this context requires clarification. Elaborate on what is meant by "robustness" and how it is measured or determined in this case, as larger standard deviation typically indicates a higher level of variability, which might be interpreted as less robustness. Providing additional context will help readers understand the intended meaning.

---

## Author Response (AR2)

**General Response to Referee #2**

Original referee comments are in blue.
Our responses are in black with regular font format. Text from the updated manuscript:

  *Appears in italics with 1⁄2 inch indentation with the removed parts exhibited with a  and red color while new text added is shown in green.*

**Summary of RC2 comments:**
**Major comment:** To discuss the limitations of the findings due to the small sample size. Additionally, to implement resampling techniques like bootstrapping to provide a more robust assessment of the statistical significance of the results. This would help in validating the study's conclusions despite the constraints posed by the sample size**.**

In fact, applying a statistical method that considers the small sample size enhances the quality of our analysis. As recommended by the reviewer, we used the bootstrap method with 50,000 resamples to improve the statistical robustness of our findings in section 4 (Environmental conditions), covering the main regime category analysis. By calculating both the mean and the standard deviation of these bootstrap sample means, we obtained a more reliable estimate of the population's central tendency and the uncertainty of our sample mean. This approach addresses potential limitations due to our original sample size, improving the confidence in generalizing our results.

Furthermore, it is important to mention that most results are very similar to those in our previous version, except for MLCIN and MLCAPE, which exhibited more significant differences.

Given this new application, the following text updates were made:
**Line 114:**

To account for the small sample size associated with the classification of the convective regime (fig. 4), we employed the bootstrap method, utilizing 50,000 samples to estimate the mean and standard deviation for each composite. These are represented by the new line and error bars in each figure in this section.

As desired by the reviewer, this sentence introduces the bootstrap method while also addressing concerns related to the small sample size and our approach to managing it.

For the conclusion, we also included one additional sentence to emphasize our resample approach:

**Line 407-409:** Unlike previous studies, we used an objective and reproducible method to identify and exclude days dominated by organized convection. Additionally, the bootstrap resampling method was employed in analyzing the environmental controls of each convective regime, permitting a robust assessment of the statistical significance of the results despite the relatively small sample size. The Deep regime is characterized by early morning moister conditions extending from the surface to the lower free troposphere.

To update the change in the mean values and uncertainties with the bootstrap method, we made the following updates in the manuscript:

**Line 238:** while the ShCu and Cong regimes show a decrease of 0.80 and 1.02 mm

**Line 244-246:** The PBL height among the convective regimes differs the most at 14 LST, where the convective boundary layer roughly coincides with the LCL or cloud base height (not shown), being about  500 m lower for the Deep ( 1535 m) than the ShCu ( 1998 m) regime.

**Note: These changes seem very large because we were incorrectly stating the height of the LCL instead of the PBL. This is now fixed, and the reported values are for the PBL-Height based on the bootstrap method.**

**Line 238:** At 14 LST, MLCAPE for the Deep regime ( 1074 J kg$^{-1}$) and Cong regime ( 986 J kg$^{-1}$) significantly exceed the value for the ShCu regime ( 558 J kg$^{-1}$). The change in MLCAPE from afternoon to early evening (MLCAPE(20 LST) - MLCAPE(14 LST)) is negative for the Deep ( -138 J kg$^{-1}$),  and positive for the Cong ( 139 J kg$^{-1}$) and  ShCu ( 644 J kg$^{-1}$) regimes.

**Line 272-273:** Conversely, the convergence of vapor and evaporation exceeds the precipitation term on Deep days, resulting in a net accumulation of 1.8 mm in column water vapor

1. To include the error propagation formula for the CONV term in the budget equation. This formula involves implicit assumptions, such as a lack of correlation among the terms ∂tCWV, ∂tLWP, EVAP, and PREC, which may not be immediately apparent. Explicitly stating these assumptions should enhance the clarity of the manuscript. Similarly, the authors should provide an explanation of how the standard error of the PREC term was obtained from the three considered data sources. Including this information will further improve the understanding of the methodology employed in the study.

To provide these clarifications, the following updates were made:

**Line 145-148**: Precipitation is obtained utilizing different sources, namely, aerosol observing system data, a tipping bucket, and a laser disdrometer (Section 2.1). We determine the mean of these sources and calculate the standard deviation from this sample mean. The water vapor convergence term is estimated using the mean composites for ∂CWV/∂t, ∂LWP/∂t, EVAP, and PREC. The standard deviation of mean convergence is estimated from the standard deviation of the mean $\partial_t CWV$, $\partial_t LWP$, EVAP, and PREC, i.e.,

$$\sigma_m = \sqrt{\sigma_m(\partial_t CWV)^2 + \sigma_m(\partial_t LWP)^2 + \sigma_m(EVAP)^2 + \sigma_m(PREC)^2}$$ . Although this formula involves implicit assumptions, such as a lack of correlation among the variables in the square root, the uncertainty in the water budget is primarily attributed to $\partial_t CWV$ (see fig. 9). Consequently, the uncertainty in the convergence term can also be roughly approximated by this term, with other terms contributing minimally.

2. The manuscript includes several speculative statements which are permissible because observations often cannot measure all factors influencing certain variables. However, it is essential to label these statements as speculative by using qualifiers like "likely" or "presumably," rather than presenting them as definitive without adequate evidence. As a reference, the following statements (lines 148-152) serve as examples: "... are uniform around the experimental site", "CWV and LWP relate to regions associated with the domain average of a cloud size scale", "our water balance is consistent for a temporal-spatial scale of ~ 1 hr and ~ 10 km."

We agree with this point, although subjective. Sometimes statements are used solely to describe results, and are specific to those results. In the discussion or conclusion sections, however, no such definitive statements are made.

For the examples mentioned in the comment, we modified the related paragraph to be more accurate with our hypothesis:

**Previous text**:
**Line 148-152**:
~~Note that we only employed local surface measurements. Surface fluxes depend primarily on the surface properties, which are uniform around the experimental site, at least up to 10 km away. Surface rain gauges and the microwave radiometer perform point measurements. However, since we are applying hourly averages for the water budget datasets and the composites average over many days, the mean precipitation, CWV, and LWP relate to regions associated with the domain average of a cloud size scale. Therefore, our water balance is consistent for a temporal-spatial scale of ~1 hr and ~10 km.~~

**Updated text**:
Note that we only employed local surface measurements and applied hourly averages for the water budget datasets. Hence, the water balance is consistent over a temporal scale of an hour and a spatial scale on the order of meters. However, surface fluxes depend primarily on surface properties, which are approximately uniform around the experimental site, at least up to a distance of a few kilometers. Precipitation, CWV, and LWP may not change significantly directly above the cloud scale (~1-10 km, from shallow to deep). Therefore, our analysis is likely to generalize well across a spatial-temporal scale of an hour and a few kilometers.

3. The text on lines 223 and 233 uses the phrase "little statistical significance." It is advisable to avoid referring to the statistical significance of results unless they have been objectively analyzed through parametric statistical tests or resampling strategies.

The text was updated as follows:
Above 3 km, the differences in moisture profile for all regimes  are minimal.

4. The authors have argued that the results remain consistent when using different rain coverage thresholds for the convection regime classification, other than the 2% threshold mentioned in the paper. While it may not be practical to present all possible analyses, mentioning this consistency in the text would add value to the discussions and support the robustness of the findings.

Thank you for the suggestion. Analyzing the manuscript structure, we believe that the best place to include further support for the 2% threshold is in the methods section. We added the following sentence in the manuscript:

**Line 177-179**: The classification essentially depends on the cloud evolution during the diurnal cycle. The threshold of 2% used to calculate rain cover is based on Zhuang et al. (2017), who manually tested several parameters. Their results indicated that shallow rain cover never exceeds 2%, a criterion we adopted in our definition of minimum rain cover for identifying congestus or deep clouds. The exclusion of any relevant early morning disturbance (06-10 LST) associated with important pre-convective activity guarantees that nighttime MCSs do not cause significant preconditioning.

5. The comment made during the previous review round about the statement "A mixed-layer parcel from the surface to 100hPa is used..." might have been unclear. To clarify, it is suggested that this statement could be rephrased as "the 100-hPa layer immediately above the surface" to provide better context and understanding.

Thank you for the clarification. This recommendation is now reflected in:

**Line 112-113**: A mixed-layer parcel  immediately above the surface to 100 hPa is used as the initial state for the parcel's ascent Stull (2016).

6. The authors have clarified that they neglect ice in their analysis; however, the manuscript still states "neglecting the time-variation of ice," which should be revised to "neglecting ice" for clarity. Additionally, the authors justify this neglect by stating that the time variation in LWP is negligible in the budget analysis. Since ice water path was not considered in the study, further clarification is needed to support this justification. It appears that the authors assume $\partial t$IWC $< \partial t$LWP in general, but this assumption is neither evident nor explicitly mentioned in the manuscript. To ensure transparency, the authors should provide a justification for this assumption within the manuscript.

**Line 137-139**: For the sake of this analysis, we neglect  ice  and express all terms in units of mm hr$^{-1}$ . As we show later (section 4.3), the results indicate a minor contribution of the liquid water term in the water budget, which supports ignoring the ice water term. Note that ice water paths are not necessarily smaller than liquid water paths; however, they still encompass values of comparable orders of magnitude.

7. In Section 2.3, it is recommended to clarify the term "rain" in "rain coverage" to specify that it refers to radar echoes exceeding a given threshold, regardless of whether the phase is ice or liquid. This will provide better context for the readers and improve the understanding of the methodology.

To provide further clarification, the text was updated as follows:

**Line 162-164:**
The rain coverage is calculated as the fraction of reflectivity pixels > 20 dBZ, regardless of whether the phase is ice or liquid (see Section 2.1). The echo-top is defined as the highest level where rain coverage is greater than 2%.

8. In the caption for Figure 6, it would be helpful to provide clarification on how the error bars for panels c and d were obtained.
Suggestion accepted. The caption was changed to:

**Figure 6**. Atmospheric conditions. (a) Potential temperature (K) and (b) water vapor mixing ratio (g/kg) at 08 LST radiosonde observations. The corresponding convective regime differences (Deep-ShCu and Deep-Cong) for potential temperature (, c) and mixing ratio ( rv, d) are also included. The error bars on panels c and d are obtained from the bootstrap standard error of the convective regimes, i.e.,

$$\sqrt{SE(Deep)^2 + SE(ShCu\ or\ Cong)^2}\ .$$

9. Line 320 and Fig. 14's caption: 2.5 J/g= 250 J/kg.

We corrected them.
2.5 J g$^{-1}$ (2500 J kg$^{-1}$)

10. The author's effort in explaining the interpretation of Zhuang et al. (2017)'s results in their response is appreciated. However, the original intent of the comment was to recommend revising the explanation within the manuscript (lines 390-393) for improved clarity and understanding.

Thanks. We found that our previous text was not clear enough to state what we wanted to say. Thus, it was modified as follows:

**Previous Version:**

**Line 390-394**: ~~Using GoAmazon2014/5 observations, Zhuang et al. (2017) reported a similar pattern of bulk wind shear during the wet season. However, their interpretation differs from ours. They suggested that strong wind shear would favor convection. Nevertheless, during the wet and transition seasons, Deep days are associated with weaker upper-level shear. Thus, they suggested that wind shear may have no impact or could even hinder convection. Here, we have shown that low-level shear has the greatest correlation with convection.~~

**Updated Version:**
Using GoAmazon2014/5 observations, Zhuang et al. (2017) reported a similar pattern of bulk wind shear during the wet season. They suggested that strong wind shear would favor convection. However, during the wet and transition seasons, Deep days are associated with weaker upper-level shear. Thus, they suggested that wind shear may have no impact or could even hinder convection. Here, our results support that strong wind shear at lower levels favors convection, while weaker shear at upper levels plays a minor role in convection. Therefore, low-level wind shear may facilitate the development of convection during the wet season.

11. Line 27: "deep convection in either..." should be "deep convection either..."

Done

12. Line 62: It is recommended to use an alternative term for "sensitivity," such as "assess correlations," to more accurately describe the intended analysis and avoid potential misinterpretations.

Done

13. Figure 1's caption: It is recommended to clarify the intended meaning of the phrase: "The dotted circle describes the S-band radar domain with available measurements.".

We replaced:

With:
The dotted circle (radius of 202 km) centered over the T1 site indicates the domain covered by S-band radar measurements.

14. Line 107: It would be helpful to clarify the assumptions made in the vertical integration up to 200 hPa, in cases where the sounding only reaches up to 8 km.

Thanks for the observation. Soundings covering up to 8 km are used in the atmospheric profile analysis (fig. 6). The calculation of CWV considers soundings extending up to 350 hPa (and not 200 as mentioned, we updated this threshold in the latest manuscript version).

**Line 104-105**: For consistency, we  analyze the atmospheric profile (potential temperature and humidity) only for soundings that extend from the surface to at least 8 km, and we linearly interpolate the profiles to a fixed 50 m vertical grid.

**Line 106-107**: The total column water vapor (CWV) is determined by integrating the water vapor mixing ratio from the surface to  350 hPa. For consistency, here we only analyze soundings that extend from the surface to at least 350 hPa (approximately 8.5 km). Similarly, the partially-integrated CWV is calculated for the layers 1000-850 hPa …

15. Line 244: The term "mixed-layer MLCIN AND MLCAPE" contains redundant elements, as the "ML" prefix already indicates that these variables refer to the mixed-layer. Consider revising this to "MLCIN AND MLCAPE" for conciseness and clarity.

Done

16. Line 261: The phrase "Changes in CWV" should be represented as $\partial t$CWV for clarity. Regarding "changes in condensate," please specify if this refers to $\partial t$LWP. Furthermore, it appears that "as" might be a more appropriate choice than "whereas," and "low" instead of "lower." Please review and revise the wording accordingly. It is important to note that negative CONV (or divergence) is not solely due to the mentioned factors, as EVAP is actually the dominant term. Please adjust the explanation to accurately reflect this information.

Yes, changes in condensate refers to $\partial t$LWP. We modified the text as suggested. Indeed, 'as' and 'low' are the right words. We have omitted evaporation from our analysis because it exhibits similar patterns across all convective regimes, despite its large values.

We replace:

With:
The daytime of ShCu and Cong days shows mostly water vapor divergence. This is primarily due to more significant negative changes in $\partial t$CWV and low precipitation rates, as evaporation shows smaller differences among the convective regimes and changes in $\partial t$LWP exert a minor influence on the water budget.

17. Figure 9: It is recommended to use a symmetric logarithmic scale for the y axis.

We are not sure if the reviewer pointed to the right figure. Using a log scale for the water budget figure is not suitable.

18. Line 282: The phrase "more significantly affected by PBL growth" can be reworded as "likely more influenced by PBL growth compared to other regimes" to enhance clarity and accurately convey the intended comparison.
Done

19. Line 285: It is recommended to add "respectively", at the end of the phrase.
Done

20. Line 309: The term "more robust" in this context requires clarification. Elaborate on what is meant by "robustness" and how it is measured or determined in this case, as larger standard deviation typically indicates a higher level of variability, which might be interpreted as less robustness. Providing additional context will help readers understand the intended meaning.

Thank you for pointing that out. As noted, the phrase needed to be clarified, given that larger uncertainty corresponds to less robustness. We updated as follows:

**Previous version**

**Updated version**
For the 0-6 km layer, the Cong and Deep regimes exhibit similar patterns, while ShCu days are characterized by larger wind shear values and greater variability.

We hope these answers provide clarifications and responses to the comments raised in this revision.

---

## Author Response (AR3)

**Response to Referee #2**

Original referee comments are in blue.
Our responses are in black with regular font format. Text from the updated manuscript:

> *Appears in italics with 1⁄2 inch indentation with the removed parts exhibited with a  and red color while new text added is shown in green.*

**RC2 comments:**
- The sentence 'Precipitation, CWV, and LWP may not change significantly directly above the cloud scale (~1-10 km, from shallow to deep)' appears somewhat unclear.

Here, we would like only to provide an order of magnitude instead of defining an accurate spatial scale. For this purpose, the sentence was updated as follows:

**Line 158-159**:
Measurements of precipitation, CWV, and LWP are influenced by the cloud cover around the instrumentation location. Given that shallow-to-deep convection typically spans a spatial scale of approximately 1-10 km, our analysis is likely to generalize well across a spatiotemporal scale of one hour and a few kilometers.

- The statement 'A mixed-layer parcel immediately above the surface to 100 hPa is used as the initial state for the parcel's ascent Stull (2016).' could be misinterpreted and needs rephrasing for better clarity. It appears to imply that the mixed layer extends from the surface to 100 hPa, which is not accurate.

The sentence was updated as follows:

**Line 113-114**:
A mixed-layer parcel immediately above the surface, extending to a depth of 100 hPa, is used as the initial state for the parcel's ascent Stull (2016).

- For Figure 9, the authors did not understand my previous suggestion to use a symmetric-logarithmic scale. More information can be found at this link: https://stackoverflow.com/questions/3305865/what-is-the-difference-between-log-and-symlog.

Thanks for your suggestion. Indeed, the symmetric log scale can be used to rescale the y-axis of Figure 9 (water budget). The resulting figure is shown below:

[Figure]

The visualization of this figure is somewhat impaired due to the presence of too much information within the same range of rate values: [-1, +1] mm/day. However, this issue persists for both the linear and symmetric log scales. Thus, the figure above does not provide an improvement over the previous figure using the linear scale.

We attempted to modify the parameter linthresh (https://matplotlib.org/3.3.3/api/_as_gen/matplotlib.colors.SymLogNorm.html) to achieve a more detailed visualization of small values in the range [-1, +1] mm/day, but the results are unsatisfactory. For example, the figure below corresponds to linthresh = 1:

[Figure]

Note that as the linthresh is reduced, the error bars will significantly affect the visualization of rates in the range of [-1, +1] mm/day.

Therefore, the symmetric log scale does not actually improve the visualization of small rates and increases the complexity of interpreting values on an uncommon scale. Thus, we believe that the original figure should be maintained in the manuscript.

We hope these answers provide clarifications and responses to the comments raised in this revision.